# Learning Rewrite-Invariant Reasoning with Targeted Alternation Training

**Mousa Arraf** [1]
**Ido Guy** [2]
**Kira Radinsky** [1]

## Abstract

Large language models (LLMs) often fail in systematic, model-specific ways under meaning-preserving question rewrites (paraphrases, format changes, benign distractors). In this work, we address this instability by identifying where the model's reasoning process diverges across semantically-equivalent inputs. For each target LLM, we sample multiple solution traces under rewrites and aggregate them into a graph of recurring intermediate steps, which pinpoints where incorrect traces diverge from correct ones. We then generate a small set of semantics-preserving examples that mirror the rewrite patterns most responsible for these divergences, and use them to steer the model (*targeted alternation training*), either via fine-tuning or via in-context learning. Across MMLU-Pro, Big-MATH, and DROP, this yields consistent gains and cross-dataset generalization. On Humanity's Last Exam, using 200 in-context examples, it improves GPT-5.2 (xhigh) from 35.4% to 38.1%, demonstrating that targeted alternation training can materially improve a frontier, API-accessible closed model under realistic access constraints.

## 1. Introduction

A small, meaning-preserving change in how a question is written can flip a strong LLM from correct to confidently wrong. For example, consider the following question: 'A recipe uses 3 cups of flour for 12 cookies. How many cups for 20 cookies?' Most models compute $3 \cdot 20/12 = 5$. Now add a benign parenthetical: 'A recipe uses 3 cups of flour for 12 cookies (the baker also buys eggs on Tuesdays).

How many cups for 20 cookies?" The correct answer is unchanged, yet many models now repeatedly fall into the same mistake: they commit early to an additive framing ('20 is 8 more than 12, so add 8') and then follow a coherent but incorrect trajectory, leading to wrong answers, such as $3+8 = 11$. This is the brittleness we study: *meaning-preserving global alternations*, e.g. paraphrases, format/style changes, verbosity shifts, benign distractors, etc., that reliably trigger the same error pattern. Such brittleness undermines reasoning reliability: rewrites can steer the model into fragile reasoning routines without making the task any harder.

A natural response is data augmentation (Wei et al., 2019): train on variants so the model learns invariances. However, most augmentation pipelines treat perturbations *uniformly* (generate many variants and add them all) (Feng et al., 2021), which inflates data and dilutes the signal with uninformative rewrites. At the other extreme, adversarial approaches (Wang et al., 2024b) search for high-loss perturbations, but the failures found are frequently idiosyncratic. The gap is conceptual: we lack a scalable method that identifies *which* alternations systematically break a model, and convert that knowledge into training data that transfers across tasks.

To close this gap, we focus on *how* a model arrives at an answer across repeated attempts, not just whether the final answer is correct. For each training question, we run the model multiple times and collect its step-by-step solution traces (intermediate reasoning steps produced during generation), optionally under a small set of meaning-preserving rewrites to broaden coverage. We then merge all traces into a *reasoning graph*: nodes represent recurring intermediate steps (after semantic merging), and edges represent transitions observed in the sampled traces. The rewrites serve only to elicit diverse traces; the core analysis and data construction are defined on the induced graph.

In the recipe example, many incorrect rewrite attempts might share the same early commitment (misframing proportionality as additivity). In the reasoning graph attributed to the question, these attempts cluster into a high-traffic region associated with the distractor rewrite, making it possible to localize *where* failure begins. Rather than treating each trace as a single correct/incorrect label, we identify the

---

[1]Technion Israel Institute of Technology, Haifa, Israel [2]Ben-Gurion University of the Negev, Beer Sheva, Israel. Correspondence to: Mousa Arraf <a.mosa@cs.technion.ac.il>.

*Proceedings of the 43rd International Conference on Machine Learning*, Seoul, South Korea. PMLR 306, 2026. Copyright 2026 by the author(s).

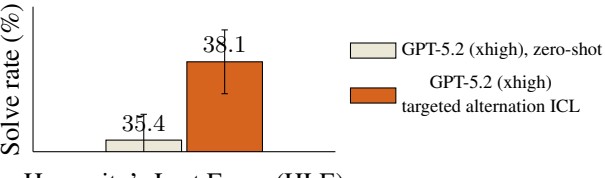

*Figure 1.* HLE (text only) solve rate for GPT-5.2 (xhigh) in zero-shot and with targeted alternation-driven 200 in-context examples. Error bars denote 95% confidence interval.

earliest divergence after which a trace can no longer reach a correct answer *within the constructed graph*. This "failure boundary" separates partially correct, still-recoverable reasoning from states that reliably lead to an incorrect outcome. We formalize this boundary as the *Solution Boundary Cut* (SBC), computed separately for each question on its per-question reasoning graph, which marks the edges where trajectories first cross from recoverable to irrecoverable states. Across many training questions, we then aggregate these per-question boundary crossings to discover the recurring failure modes and rewrite patterns that most often trigger them. This turns a global objective (e.g., accuracy) into a set of local, interpretable transition events, allowing us to ask: which recurring patterns identify wrong answers?

Using failure patterns that recur across many training questions, we synthesize a small set of targeted, semantics-preserving training instances that isolate the rewrite cues most responsible for errors. Concretely, for each question we *extract a shared prefix and* construct an *alternation pair*: two meaning-preserving rewrites that follow the same prefix but then diverge, with one leading to an incorrect answer (exhibiting the failure pattern) and the other to the correct answer. We apply these pairs via fine-tuning (open-weight models) or as in-context demonstrations (closed models), and call the resulting approach *targeted alternation training*.

**Results.** Across MMLU-Pro, Big-MATH, and DROP, targeted alternation training improves accuracy and rewrite robustness relative to supervised fine-tuning, uncertainty-aware fine-tuning, augmented question finetuning, and test-time ensembling. On Humanity's Last Exam (HLE), we obtain a reliable lift for a closed frontier model using in-context targeted alternations: 200 examples raise GPT-5.2 (xhigh) solve rate from 35.4% to 38.1%[1] (Figure 1), which is competitive with the best publicly reported text-only results.

**Contributions.** (i) We propose an end-to-end procedure that perturbs questions with rewrites, samples multiple solution traces, and aggregates them into a *reasoning graph* of recurring intermediate steps. (ii) We introduce SBC, a

graph-derived failure boundary that localizes where traces enter regions that reliably end in incorrect answers; we use it to identify rewrite families associated with these failures and construct targeted alternation examples. (iii) We show consistent gains and cross-dataset transfer on MMLU-Pro, Big-MATH, and DROP, and achieve competitive results with SOTA on Humanity's Last Exam via targeted alternation in-context learning. Code and data available here [2].

## 2. Related Work

Many robustness methods improve invariance to surface form by training on semantics-preserving perturbations, including edit-based substitutions (Wei et al., 2019), back-translation (Li et al., 2024), and counterfactual rewrites (Plyler & Chi, 2025). Related work also includes adversarial or robustness-regularized objectives that encourage stability under perturbations (e.g., TextAttack, FreeLB, SMART) (Morris et al., 2020; Zhu et al., 2020; Jiang et al., 2020). A complementary strategy is to focus training on the hardest perturbed instances, often using uncertainty or margin-based selection from an augmented pool (Yang et al., 2025). Step-DPO (Xu et al., 2025) provides local step-level supervision by preferring a better next step over a worse one. In contrast, SBC identifies the transition from recoverable to unrecoverable reasoning and trains on those boundary cases. Thus, Step-DPO improves step quality, while SBC targets robustness to reasoning collapse.

Finally, parameter-free test-time techniques such as self-consistency (Wang et al., 2023a) and sampling-and-voting aggregate multiple stochastic executions or reformulations to reduce variance (Wang et al., 2023b). We evaluate representatives of these categories as baselines.

Our approach differs in mechanism: we use a small set of meaning-preserving rewrites to identify systematic errors, summarize repeated runs as a per-question graph, identify the rewrite-induced step where solutions start going wrong, and train on compact contrastive rewrite pairs using preference optimization (Rafailov et al., 2023).

## 3. Reasoning Graph Construction

Our goal is to make a target LLM more robust to meaning-preserving rewrites. We begin by building an explicit, per-question representation of how the model reasons across repeated attempts. For each base question, we apply a small set of controlled rewrites and sample multiple step-by-step solution traces from the target model (Section 3.1). We then canonicalize each trace into comparable steps (Section 3.2) and merge recurring steps across runs into an empirical *reasoning graph* (Section 3.3), where nodes correspond to in-

---

[1]Statistical significance achieved using McNemar's test on paired per-question outcomes confirming that the improvement is statistically significant (p < 0.05)

[2]https://github.com/arrafmousa/SBC

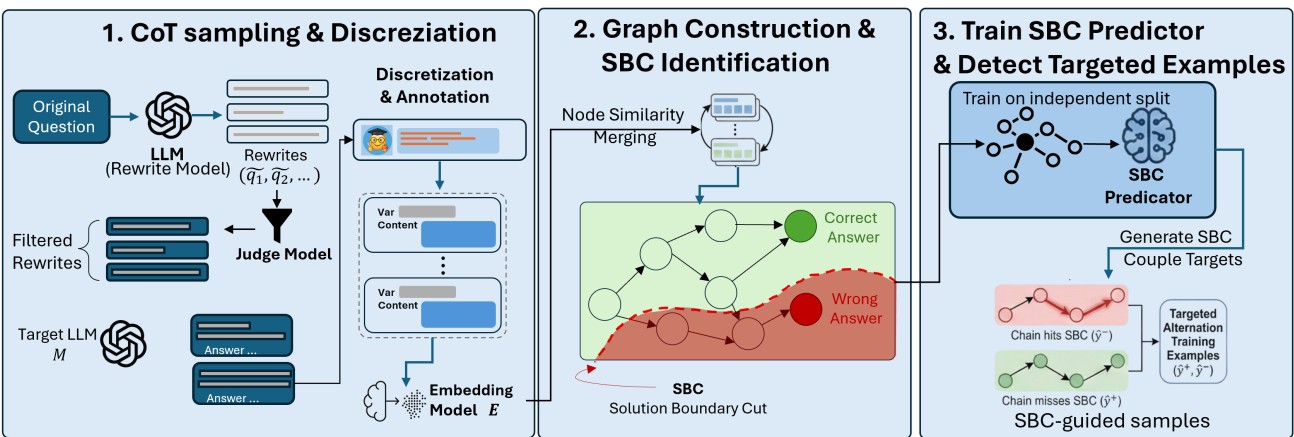

*Figure 2.* Overview of targeted alternation pipeline. (1) generate meaning-preserving rewrites, sample solution traces from target model $M$, and discretize them into intermediate steps. (2) merge similar steps into a reasoning graph; identify transitions from recoverable states to irrecoverable ones, forming the SBC. (3) train an edge-level predictor on independent graphs and use high-confidence boundary crossings to construct targeted success/failure alternation pairs.

termediate steps and directed edges correspond to observed step-to-step transitions. This graph is purely an aggregated view of the model's observed trajectories under stochastic decoding and rewrites; it serves as the substrate for identifying recurring failure structure in Section 4.

### 3.1. Sampling Solution Traces

Given a base question $q$, we generate up to $R$ rewrite candidates $\{\tilde{q}\}$ using $T$ fixed prompt templates (Appendix C) covering paraphrasing, format/style shifts, syntactic restructuring, and benign distractors—perturbation classes known to affect LLM performance despite preserving intent (Ribeiro et al., 2020; Jia et al., 2017).

We retain only rewrites judged semantically equivalent to $q$ by an independent judge model (GPT-5-MINI), which rejects 2%. The judge is used only for equivalence filtering and step annotation (Sections 3.2–3.3), not to select answers. We audit the equivalence filter via random human spot-checks; details appear in Appendix D.

For each retained rewrite $\tilde{q}$ (and the original $q$), we sample $K=2$ *solution traces* from the target model $M$ using stochastic decoding (default hyper-parameters as published by model authors). We use K=2 as a practical sampling budget, with sensitivity analysis in Appendix G.1 showing diminishing changes beyond K=2–3. Traces are elicited with a fixed step-by-step instruction prompt in the spirit of chain-of-thought prompting (Wei et al., 2022): *"You are an expert AI assistant that explains your reasoning step by step."* A *solution trace* is the generated step-by-step text. We treat it only as an observable textual artifact of the decoding process and do not assume it is a faithful account of the model's internal computation. Section 3.2 canonicalizes these traces into discrete steps that can be compared and

merged across runs for graph construction.

### 3.2. Discretizing and Annotating Solution Traces

The sampled solution traces are free-form text and vary in formatting and granularity across runs. To enable consistent comparison across runs and rewrites, we canonicalize each trace into a standardized sequence of steps. Specifically, we use a small helper model (GPT-5-MINI) to insert explicit step delimiters and split the trace into an ordered list of short spans, each intended to capture a single intermediate decision or update. This step is purely structural: the helper model is instructed to only insert delimiters and preserve the original wording. We verify preservation by removing delimiters and normalizing whitespace, then requiring the normalized Levenshtein distance between the reconstructed trace and the original trace to be $\leq 0.05$; we additionally audit 100 random traces with human spot-checks (Appendix E).

We then attach lightweight semantic annotations to each step, using GPT-5-MINI conditioned on the original question, the full trace, and the final answer. For each step, the annotator outputs: (1) a *slot/variable tag* describing the main quantity or fact the step is about (e.g., SCALE_FACTOR, INTERMEDIATE_VALUE); and (2) an *operation label* from a small fixed set. We use four coarse operation types: RE-TRIEVE (restating or extracting information present in the input/context), INFER (deriving a new statement from previously established information, e.g., arithmetic or logical composition), GENERATED (introducing background knowledge not explicit in the input), and OTHER. Finally, we embed each step using an embedding model $E$ (step text and slot/variable tag). We use BGE-Large-EN (Xiao et al., 2023) as the embedding model in this step. These represen-

tations, together with the operation label, are used to decide when two steps from different traces should be merged into the same graph node (Section 3.3). Prompt details are provided in Appendix E. We emphasize that discretization is used as a post-hoc probe to reverse-engineer model behavior from black-box outputs, rather than as a claim about internal representations.

Coming back to the recipe example, a correct trace elicited under a *format-shift* rewrite (e.g., a rewrite asking to present the quantities as a short bullet list) might be discretized into:

> **[Step 1 | RETRIEVE]** Extract quantities: 3 cups for 12 cookies; query is 20 cookies.
> **[Step 2 | INFER] Proportional framing**: scale by $20/12$.
> **[Step 3 | INFER]** Compute $3 \cdot (20/12) = 5$.
> **[Final | RETRIEVE]** 5

In contrast, a trace elicited under a *benign-distractor* rewrite (e.g., buying the eggs on Tuesday) may trigger an early additive commitment:

> **[Step 1 | RETRIEVE]** Extract quantities: 3 cups for 12 cookies; query is 20 cookies.
> **[Step 2| INFER] Additive framing**: 20 is 8 more than 12, so add 8.
> **[Step 3 | INFER]** Compute $3 + 8 = 11$.
> **[Final | RETRIEVE]** 11

Discretization localizes divergence to the step level (Step 2), enabling aggregation across runs.

### 3.3. Reasoning-Graph Construction

For each base question $q$, we aggregate the canonicalized, step-annotated traces sampled from $q$ and its accepted rewrites $\{\tilde{q}\}$ into a directed reasoning graph $G_q = (V_q, E_q)$. Each step is represented by (i) an embedding of the step text, (ii) an embedding of its slot/variable tag (Section 3.2), and (iii) a coarse operation type. We map a step to an existing node if it matches a previously created node under our merge rule (same operation type and weighted cosine similarity—over the two embeddings—exceeding a threshold); otherwise we create a new node. Weights and thresholds are tuned on a validation split; see Appendix I.1.

Given a trace, consecutive mapped steps induce a directed edge; we weight each edge by the number of times the corresponding transition is observed across traces. Because semantically similar steps can be reused across traces (and, in rare cases, within a trace), $G_q$ may contain cycles; we therefore treat it as an empirical transition graph over observed trajectories rather than a tree or DAG. To avoid quadratic comparisons during node lookup, we index existing step nodes using HNSW approximate nearest-neighbor search over step embeddings, and only apply the merge rule to the retrieved candidates; details are provided in Appendix I.2.

In the recipe question, the quantity extraction step typically maps to a shared node across traces, while the proportional vs. additive framing steps map to distinct nodes, yielding a shared prefix that branches at the framing decision. We ablate the use of these meta-variables in Appendix G.2.

## 4. Targeted Alternation Training

The reasoning graph from Section 3.3 provides an explicit, per-question view of the target model's observed intermediate states and transitions (aggregated over rewrites and stochastic runs). Here we use this graph to (i) localize where trajectories become irrecoverable under the constructed reasoning graph, and (ii) convert these localized failure signals into a small set of semantics-preserving intervention examples, which we call *Targeted Alternation Examples*, that improve rewrite robustness. Concretely, for each base question $q$ and target model $M$, we compute a *Solution Boundary Cut* (SBC) by marking which graph states can still reach a terminal node matching the gold answer, and identifying transitions that cross into states that cannot. To generalize beyond a single question, we train a model-specific *boundary-crossing predictor* that learns cross-question patterns associated with these crossings, and use its high-confidence predictions to select and construct the alternation examples. We apply the resulting examples via fine-tuning when available or as in-context demonstrations for closed models; we refer to this overall pipeline as *targeted alternation training*. The complete training process is illustrated in Figure 2.

### 4.1. Solution Boundary Cut

The per-question reasoning graph $G_q$ contains both successful and failed trace fragments. We use it to localize *where* a trajectory enters regions from which a correct answer is no longer reachable under the reasoning graph.

**Correct terminals.** Let $G_q = (V_q, E_q)$ be the reasoning graph for question $q$. Let $T_{\text{corr}} \subseteq V_q$ denote the set of terminal nodes whose final answer matches the gold answer for $q$ under the task's evaluation rule (e.g., exact match, numeric tolerance, or DROP normalization). In the recipe example, the reasoning graph contains two continuations after the shared quantity-extraction node: one that commits to a proportional framing and reaches a correct terminal (5), and one that commits to an additive framing and reaches an incorrect terminal (11). Here the set of correct terminals $T_{\text{corr}}$ consists of terminal nodes whose final answer is 5.

**Recoverable vs. irrecoverable states.** A node $v \in V_q$ is *recoverable* if there exists a directed path in $G_q$ from $v$ to some $t \in T_{\text{corr}}$; otherwise $v$ is *irrecoverable*. This irrecoverability is an observational notion: it indicates that, under the set of continuations explored in $G_q$, the model did not

recover from the error. As additional traces are sampled, constructed reasoning graph monotonically refines this approximation. In the recipe graph, the shared prefix nodes (quantity extraction) are recoverable because each has a path to the correct terminal 5. In contrast, the additive framing node is irrecoverable under the observed transitions: all sampled continuations from it terminate in an incorrect answer and no path reaches $T_{\text{corr}}$.

**Definition 4.1** (Solution Boundary Cut). Let $V_{rec} \subseteq V_q$ be the set of recoverable nodes and let $V_{irr} = V_q \setminus V_{rec}$ be the irrecoverable nodes. The SBC is the partition $(V_{rec}, V_{irr})$ together with the set of boundary-crossing edges $E_{\text{SBC}} = \{(u, v) \in E_q \mid u \in V_{rec}, v \in V_{irr}\}$, i.e., transitions that move from recoverable to irrecoverable regions.

In the recipe example, the edge from the quantity-extraction node to the additive framing node is an SBC edge: it leaves the recoverable region (exists a path to 5) to the irrecoverable region (no path to 5 exists within the constructed graph).

**Computation.** We compute the recoverable set $V_{rec}$ by a standard graph traversal (BFS) on the reversed graph, seeded at the correct terminal nodes $T_{\text{corr}}$. All nodes reached in this reverse traversal are marked recoverable; the remainder are irrecoverable, $V_{irr}$. We then obtain the boundary-crossing edges with a single pass over edges, $E_{\text{SBC}}$. This takes $\mathcal{O}(|V_q| + |E_q|)$ time and is valid even when $G_q$ contains cycles (Algorithm 1). In the recipe example, seeding the traversal at the correct terminal 5 marks the proportional-framing state and the shared quantity-extraction prefix as recoverable, while leaving the additive-framing state unmarked (irrecoverable). The scan therefore places the transition from quantity extraction $\rightarrow$ additive framing in $E_{\text{SBC}}$.

**Earliest boundary crossing.** Each sampled solution trace corresponds to a walk in $G_q$. We define its *first SBC crossing* as the earliest step index at which the walk traverses an edge in $E_{\text{SBC}}$. In Section 4.3, we use these first-crossing locations to isolate the rewrite-induced decision that most directly precedes irrecoverable failure and to construct the alternation examples.

### 4.2. Boundary-Crossing Predictor

The SBC is computed within each per-question reasoning graph $G_q$, but we ultimately want a signal that generalizes across many questions so we can select rewrite patterns that tend to trigger irrecoverable failures for a given target model. To do so, we learn a *boundary-crossing predictor* that estimates whether an edge is likely to cross from recoverable to irrecoverable states. The boundary crossing in the recipe example (shared prefix $\rightarrow$ additive framing) might recur in other rate-scaling questions, e.g., "A car travels 10 miles in 30 minutes. How long for 16 miles?" Under

a benign-distractor rewrite, some traces commit to a difference/additive move ("16 is 6 more than 10, so add 6 minutes"), which becomes irrecoverable in the per-question reasoning graph (no path reaches the correct terminal). Across many training questions (ratios, speed/time, work-rate), these early "difference framing" transitions share a common structural/semantic signature, which the boundary-crossing predictor learns to score as likely recoverable to irrecoverable crossings.

**Learning setup.** For a fixed target model $M$, we construct graphs on a training split, compute the ground-truth SBC per question (Section 4.1), and label each directed edge $(u, v) \in E_q$ as:

$$y(u, v) = 1_{uv}[u \in V_{rec}, \ v \in V_{irr}]$$

i.e., whether it is an SBC edge. We train the predictor only on this disjoint split and never use it to select answers or score model outputs.

**Features and model.** We represent each directed edge (u,v) with lightweight features that summarize how frequently the transition is taken in sampled traces and where it occurs in the graph, including transition counts or empirical transition probabilities, minimum distance from any root node, minimum distance to a terminal node, and local branching statistics such as out-degree and proximity to junctions. Full feature definitions are provided in Appendix F. In our main experiments, the boundary-crossing predictor is a supervised GraphSAGE-based edge classifier. Node representations are computed with a message-passing encoder using residual connections and LayerNorm. For each directed edge (u,v), an MLP receives the concatenation of the source node representation, destination node representation, and raw edge features, and predicts whether the edge crosses from a recoverable to an irrecoverable region. At inference time, we apply a min-cut decoder to select high-confidence boundary-crossing edges used for targeted alternation construction. Alternative predictor architectures are evaluated in Section 7.2.

**Usage.** At inference time, on the training questions used to construct the targeted alternation examples, the predictor assigns a score to each edge and identifies a set of high-confidence boundary crossings, denoted $\hat{E}_{\text{SBC}}$. We use these predicted crossings to identify solution paths that exhibit failure patterns typical of irrecoverable errors, and to select the corresponding targeted alternation examples for model fine-tuning or in-context learning.

### 4.3. Constructing Targeted Alternation Examples

For each question $q$, we construct an *alternation pair* by first selecting a high-confidence predicted boundary edge

$(u, v) \in \hat{E}_{\text{SBC}}$ for which there exist at least two traces that reach node $u$ and diverge immediately afterward, with one trace taking a predicted boundary edge and the other avoiding all predicted boundary edges. The shared prefix of these traces up to $u$ defines a common trace prefix $\pi_u$.

We then search over accepted rewrites of $q$ and sample traces until we identify: (i) a *success-inducing* rewrite $\tilde{q}^+$ whose trace follows $\pi_u$ and does not cross the predicted SBC, and (ii) a *failure-inducing* rewrite $\tilde{q}^-$ whose trace follows $\pi_u$ and then crosses the predicted SBC. If multiple candidates exist, we select the pair with the longest shared prefix.

The resulting two training examples are $x^+ = (\pi_u, \tilde{q}^+)$ and $x^- = (\pi_u, \tilde{q}^-)$, serialized either as fine-tuning instances using DPO optimization or as in-context demonstrations shown side-by-side to contrast the two outcomes. We call this SBC-guided finetuning.

# 5. Experimental Setup

We evaluate targeted alternation training across open- and closed-weight LLMs and multiple benchmarks, using fine-tuning or in-context learning depending on access. Our design isolates the effect of constructing targeted alternation examples, while controlling for dataset- and model-specific artifacts via cross-dataset training.

## 5.1. Models

We study targeted alternation training under two realistic access regimes. For trainable settings, we use four models where parameter updates are feasible in our environment—LLaMA 3.3 70B, GPT-4.1-mini, Phi-4, and DeepSeek-V2-lite —chosen to span open-weight (or fine-tuneable) families and a range of sizes/architectures. For the closed-weight regime, we evaluate GPT-5.2, which is API-accessible but not fine-tunable in our setting; for this we apply targeted alternation training only via in-context.

## 5.2. Datasets

We construct per-question reasoning graphs and targeted alternation examples on three reasoning benchmarks: MMLU-PRO (Wang et al., 2024a), Big-MATH (Albalak et al., 2025), and DROP (Dua et al., 2019). We chose these datasets to cover complementary reasoning regimes—broad multi-domain problem solving (MMLU-PRO), multi-step symbolic/numeric computation (Big-MATH), and discrete reasoning over passages (DROP). We apply a unified preprocessing pipeline and benchmark-appropriate exact-match-style scoring. We additionally report results on Humanity's Last Exam (HLE) (Phan et al., 2025) as a frontier *evaluation-only* benchmark. Because open-weight models in our study achieve near-zero accuracy on HLE (limiting meaningful per-question graph structure) and GPT-5.2 is near-saturated

on MMLU-PRO/Big-MATH/DROP, we learn targeted alternation examples on the three source benchmarks and separately test transfer to HLE via in-context learning.

## 5.3. Training Protocol

For trainable models, we follow Section 4: for each source-benchmark training question, we generate five accepted meaning-preserving rewrites and sample K=2 traces for the original question and each rewrite, yielding 12 sampled traces per base question before graph construction. We then build a per-question reasoning graph and construct targeted alternation preference pairs $(y^+, y^-)$ that share a prefix but diverge immediately after it, with one continuation reaching a correct terminal and the other exhibiting the failure pattern. Across the source benchmarks, the final SBC construction set contains approximately 60K structured trace instances and 17.2M generated/annotated tokens, distributed roughly evenly across MMLU-PRO, Big-MATH, and DROP. We fine-tune with DPO (Rafailov et al., 2023), using each model's published default fine-tuning hyperparameters. To assess transfer and avoid leakage, we use leave-one-benchmark-out training on {MMLU-PRO, Big-MATH, DROP}: when evaluating on a target benchmark, preference pairs are constructed only from the training splits of the other two benchmarks, and the target test split is never used for trace sampling, graph construction, pair selection, or hyperparameter choices. For GPT-5.2 (xhigh), we cannot update parameters; we therefore convert preference pairs into ordered demonstrations $(q, y^+, y^-)$ and prepend 200 demonstrations, limited by the context window. Evaluation is performed on the original, unaugmented test questions using each benchmark's exact-match-style scoring.

## 5.4. Baselines

We compare targeted alternation training to baselines that isolate five competing sources of robustness: (i) *standard task supervision* via supervised fine-tuning on gold question-answer pairs, without reasoning-graph construction or preference pairing; (ii) *Random rewrite pairing*: DPO fine-tuning on randomly matched success/failure pairs constructed from meaning-preserving rewrites drawn from the same rewrite generator and equivalence filter as our pipeline, but without boundary localization or prefix-controlled pairing (iii) *hard-example mining on rewrites* via uncertainty-based selection (logit-margin) (Yang et al., 2025): from the same rewrite-augmented pool, prioritize rewrite instances where the model is least confident under a logit-margin uncertainty proxy, then fine-tune with the same optimizer and compute budget; (iv) *inference-time rewrite ensembling* via test-time self-consistency over rewrites (Wang et al., 2023b), i.e., answer multiple meaning-preserving rewrites and aggregate predictions by majority vote, without param-

eter updates; and (v) *positive-only trajectory learning* using the same pipeline and training budget as targeted alternation training, but fine-tuning only on trajectories that end in the correct answer (no matched negative/failure example at a shared prefix). For all training-time baselines, we match our training protocol and compute budget (same splits, the same rewrite generator/equivalence filter when applicable, and the same optimizer/training schedule); methods differ only in how training instances are selected or labeled.

# 6. Empirical Results

We evaluate SBC-guided finetuning across multiple models and reasoning benchmarks, focusing on both accuracy improvements and cross-dataset generalization. All reported results follow the cross-dataset protocol described in Section 5, where models are trained on SBC-derived failure pairs from datasets disjoint from the evaluation set.

## 6.1. Main Results

Table 1 reports accuracy on Big-MATH, MMLU-PRO, and DROP across four model families. Overall, targeted alternation training yields the most consistent gains across models and datasets relative to standard fine-tuning and rewrite-based baselines.

Answer-only supervised fine-tuning (SFT; gold question–answer supervision) is not consistently beneficial under a fixed training budget: it sometimes provides small improvements but can also be neutral or slightly negative, suggesting that rewrite robustness is not obtained simply by additional QA supervision. Random rewrite augmentation and uncertainty-based hard-example mining on rewrites (logit-margin) improve robustness in some settings, but their gains are typically smaller than targeted alternation training. Test-time self-consistency over rewrites (majority vote) provides moderate improvements without parameter updates, and can be competitive on MMLU-PRO, but it is generally weaker on multi-step benchmarks.

The largest improvements from targeted alternation training appear on Big-MATH and DROP, which emphasize multi-step computation and stateful reasoning. Finally, the positive-only trajectory learning ablation (training only on correct trajectories, without matched failure pairs at a shared prefix) improves over SFT but underperforms the full targeted alternation setup, indicating that contrastive pairing is important beyond simply reinforcing correct solutions.

The drop observed for DeepSeek-V2-Lite under standard SFT is concentrated in that backbone rather than appearing uniformly across models, even though the same SFT and evaluation pipeline was used throughout. This suggests model-specific negative transfer rather than a systematic implementation artifact. One plausible factor is that DeepSeek-

V2-Lite is an MoE model with only 2.4B active parameters per token despite 15.7B total parameters, which may make it more sensitive to final-answer-only SFT. This is consistent with prior work showing that SFT can induce ability interference, overfitting, catastrophic forgetting, or benchmark regressions in some models (Fu et al., 2024; Gupta et al., 2025), and that response-/answer-only training can reduce reasoning robustness (Chatterjee et al., 2025).

## 6.2. Targeted Alternation In-Context Learning on HLE

We further evaluate targeted alternation in a frontier, closed-weight setting on HLE (text-only). Scale's public text-only leaderboard reports a top score of 37.72 ± 2.04 (Gemini-3-Pro-Preview) under third-party evaluation, but this system is not accessible in our setup via a programmatic evaluation API. We therefore evaluate GPT-5.2 (xhigh), a strong foundation model, and apply targeted alternation via in-context learning, without parameter updates. We convert each targeted alternation pair into a contrastive demonstration $(q, y^+, y^-)$ and include up to 200 demonstrations per prompt, which is the practical limit imposed by the context window.

On the text-only subset of approximately 2,000 HLE questions, targeted alternation prompting improves GPT-5.2 (xhigh) from 35.4% to 38.1%. The paired zero-shot and targeted-alternation outcomes are statistically significant under McNemar's test ($p < 0.05$). We do not claim state-of-the-art across all settings, since public reports often differ in model access, tool use, and evaluation protocol. Instead, we use the Scale leaderboard as a third-party reference point and show that targeted alternation yields a reliable lift for a strong, API-accessible closed model.

# 7. Ablations

## 7.1. Effect of Meaning-Preserving Alternations

To isolate the contribution of meaning-preserving alternations, we ablate our training pipeline by removing rewrites. In the *no-alternations* setting, we build each per-question reasoning graph from multiple sampled traces of the original question only (no rewrites), while matching the total number of sampled traces to the alternation-enabled setting. We then construct preference pairs using the same graph-to-pair procedure and fine-tune with the same DPO budget. Table 2 reports results for GPT-4.1-mini. Without alternations, we do not observe consistent gains over the base model, and differences are not statistically significant under a paired bootstrap test. In contrast, enabling alternations yields larger gains across all three benchmarks. We hypothesize that alternations increase the diversity of observed trajectories in the per-question graphs (i.e., more distinct next-step continuations), in part by revealing when

*Table 1.* Accuracy (%) with *estimated* 95% CIs that incorporate both evaluation-set noise and method-dependent run-to-run variance (seed/sampling effects). *Closed-weight model; uncertainty estimated from the model's own output at inference time.

| Model | Training | Big-Math | MMLU-PRO | DROP |
|---|---|---|---|---|
| DEEPSEEK-V2-LITE | Base (no training) | $52.7 \pm 0.61$ | $60.1 \pm 1.04$ | $48.1 \pm 1.15$ |
| | Standard task supervision | $54.3 \pm 0.61$ | $53.8 \pm 1.19$ | $49.8 \pm 1.28$ |
| | Hard-example mining | $54.4 \pm 0.61$ | $60.3 \pm 1.34$ | $50.6 \pm 1.43$ |
| | Random rewrite exposure | $54.6 \pm 0.62$ | $54.5 \pm 1.27$ | $50.0 \pm 1.35$ |
| | Inference-time rewrite ensembling | $53.7 \pm 0.61$ | $60.6 \pm 1.43$ | $51.2 \pm 1.51$ |
| | Positive-only trajectory learning | $56.0 \pm 0.60$ | $59.6 \pm 1.26$ | $53.1 \pm 1.35$ |
| | Targeted alternation training | $\mathbf{58.1} \pm \mathbf{0.59}$ | $\mathbf{61.8} \pm \mathbf{1.26}$ | $\mathbf{54.4} \pm \mathbf{1.35}$ |
| PHI-4 | Base (no training) | $63.4 \pm 0.58$ | $71.1 \pm 0.96$ | $76.8 \pm 0.98$ |
| | Standard task supervision | $65.0 \pm 0.58$ | $70.2 \pm 1.11$ | $76.4 \pm 1.10$ |
| | Hard-example mining | $65.1 \pm 0.59$ | $70.5 \pm 1.26$ | $75.7 \pm 1.23$ |
| | Random rewrite exposure | $65.0 \pm 0.51$ | $70.8 \pm 1.19$ | $75.0 \pm 1.19$ |
| | Inference-time rewrite ensembling | $64.2 \pm 0.587$ | $71.5 \pm 1.34$ | $82.2 \pm 1.26$ |
| | Positive-only trajectory learning | $67.2 \pm 0.57$ | $71.8 \pm 1.19$ | $81.4 \pm 1.23$ |
| | Targeted alternation training | $\mathbf{68.2} \pm \mathbf{0.56}$ | $\mathbf{72.3} \pm \mathbf{1.18}$ | $\mathbf{84.2} \pm \mathbf{1.15}$ |
| LLAMA-3.3-70B-INSTRUCT | Base (no training) | $61.5 \pm 0.58$ | $71.9 \pm 0.96$ | $67.1 \pm 1.07$ |
| | Standard task supervision | $62.0 \pm 0.57$ | $72.1 \pm 1.11$ | $67.9 \pm 1.13$ |
| | Hard-example mining | $62.2 \pm 0.59$ | $69.7 \pm 1.28$ | $71.5 \pm 1.20$ |
| | Random rewrite exposure | $62.3 \pm 0.59$ | $69.6 \pm 1.28$ | $69.0 \pm 1.16$ |
| | Inference-time rewrite ensembling | $61.9 \pm 0.52$ | $\mathbf{73.0} \pm \mathbf{1.33}$ | $74.1 \pm 1.26$ |
| | Positive-only trajectory learning | $64.4 \pm 0.53$ | $71.8 \pm 1.19$ | $74.2 \pm 1.28$ |
| | Targeted alternation training | $\mathbf{65.3} \pm \mathbf{0.58}$ | $72.8 \pm 1.18$ | $\mathbf{76.5} \pm \mathbf{1.24}$ |
| GPT-4.1-MINI-2025-04-14 | base (no training) | $67.1 \pm 0.51$ | $76.5 \pm 0.91$ | $85.9 \pm 0.87$ |
| | standard task supervision | $69.9 \pm 0.56$ | $76.7 \pm 1.05$ | $86.4 \pm 0.96$ |
| | hard-example mining* | $70.0 \pm 0.56$ | $76.7 \pm 1.27$ | $86.5 \pm 1.23$ |
| | random rewrite exposure | $70.5 \pm 0.56$ | $77.2 \pm 1.18$ | $86.0 \pm 1.14$ |
| | inference-time rewrite ensembling | $69.1 \pm 0.57$ | $77.0 \pm 1.36$ | $85.0 \pm 1.34$ |
| | positive-only trajectory learning | $70.5 \pm 0.56$ | $77.8 \pm 1.17$ | $90.1 \pm 1.09$ |
| | Targeted alternation training | $\mathbf{74.8} \pm \mathbf{0.53}$ | $\mathbf{78.0} \pm \mathbf{0.86}$ | $\mathbf{91.9} \pm \mathbf{0.89}$ |

*Table 2.* Accuracy of GPT-4.1-mini with and without generating question alternations.

| Alternation | Big-Math | MMLU-PRO | DROP |
|---|---|---|---|
| Baseline | $67.1 \pm 0.51$ | $76.5 \pm 0.91$ | $85.9 \pm 0.87$ |
| Without Alt. | $66.7 \pm 0.55$ | $76.6 \pm 0.89$ | $86.1 \pm 0.89$ |
| With Alt. (our) | $\mathbf{74.8} \pm \mathbf{0.53}$ | $\mathbf{78.0} \pm \mathbf{0.86}$ | $\mathbf{91.9} \pm \mathbf{0.89}$ |

*Table 3.* AUC (95% CI) for boundary-crossing predictors.

| Pred. | DeepSeek | GPT-4.1 | Llama-3.3 | Phi-4 |
|---|---|---|---|---|
| Log. Reg. | $70.43 \pm 3.92$ | $70.59 \pm 2.83$ | $69.86 \pm 2.87$ | $72.26 \pm 4.24$ |
| G. Trans. | $75.41 \pm 4.38$ | $74.62 \pm 2.96$ | $73.70 \pm 2.88$ | $77.20 \pm 4.41$ |
| GNN | $80.17 \pm 4.07$ | $78.12 \pm 2.76$ | $76.85 \pm 2.89$ | $83.83 \pm 3.84$ |

the model follows spurious surface cues ("noise") instead of the invariant problem structure.

### 7.2. Boundary-Crossing Predictor Architecture

We ablate the architecture used for the boundary-crossing predictor (Section 4.2), which scores whether a directed transition in a per-question reasoning graph is a boundary-crossing edge. We compare a linear edge classifier (logistic regression on hand-crafted edge features), a higher-capacity graph transformer, and a message-passing GNN that leverages local graph neighborhoods. We evaluate each predictor by its AUC in separating boundary-crossing edges from non-boundary edges on held-out questions; results appear in Table 3. Across source models, the GNN achieves the highest AUC, suggesting that incorporating graph structure

(beyond per-edge features) improves generalization. We emphasize that the predictor is used to prioritize *frequent* boundary-crossing patterns for constructing targeted alternation pairs; it need not perfectly classify every failure edge, especially for rare or idiosyncratic errors. We ablate the contribution of these predictor annotations to the overall pipeline accuracy in Appendix G.3.

## 8. Analysis of Solution-Boundary Crossings

In this section, we provide a descriptive analysis of the boundary-crossing transitions identified from the reasoning graphs and edge predictor. We focus on two complementary views: (i) which local graph properties are most associated with predicted boundary crossings (Section 8.1), and (ii) which rewrite families most frequently elicit traces that traverse these crossings (Section 8.2). This analysis is meant

to characterize recurring failure structure across models, not to introduce additional training components.

## 8.1. Structural Signatures of Boundary Crossings

We study which local graph properties are most associated with solution-boundary crossings in our per-question reasoning graphs. Using the learned boundary-crossing predictor, we score each directed edge and treat high-confidence crossings as *SBC edges*. Here, a boundary crossing is an edge from a recoverable node (can reach a correct terminal in the constructed graph) to an irrecoverable node. To summarize feature differences, we report the mean shift

$$\Delta = \mu_{\text{SBC}} - \mu_{\text{non-SBC}},$$

where $\Delta > 0$ indicates that a feature is more prevalent on SBC edges.

Across models, predicted boundary-crossing edges concentrate at "decision points": they originate disproportionately from branching nodes (higher out-degree; $\Delta \in [+0.96, +0.98]$ across models) and tend to follow low-probability continuations from those nodes (transition probability $\Delta \in [-0.52, -0.56]$ relative to typical outgoing edges from the same source). This suggests that many irrecoverable failures occur when the model reaches a choice point and commits to a weakly supported next step rather than the dominant continuation.

We also observe a "non-reconvergence" signature after the crossing. SBC edges are less likely to enter high-traffic merge regions (mean shift $\Delta \in [-0.16, -0.27]$), and trajectories that cross them exhibit reduced overlap with other sampled traces (mean shift $\Delta \in [-0.18, -0.34]$), consistent with divergence into more isolated failure paths. Finally, SBC edges tend to occur early in the trajectory: our normalized position feature is defined on $[0, 1]$ with 0 denoting the first step and 1 denoting the final step, and SBC edges have lower normalized position on average (mean shift $\Delta \in [-0.11, -0.17]$), indicating that boundary crossings relatively occur early. Overall, these statistics are descriptive: the predictor highlights *recurring* boundary-crossing transitions in the sampled trace distribution, and is not intended to account for every individual error.

Secondary feature shifts vary by model family; we report the per-model breakdown in Appendix B.

## 8.2. Which Rewrite Families Elicit Boundary Crossings

We next ask which rewrite families most often produce traces that traverse a predicted boundary-crossing edge (Table 4). Across all model families, *adding irrelevant information* is the most consistently failure-inducing: even benign distractors frequently steer trajectories into irrecoverable regions of the per-question graph. DEEPSEEK and PHI-4

show additional sensitivity to *question reordering* and *subordinate reversing*, which perturb discourse structure (e.g., inverting cause/effect or antecedent/consequent) without changing semantics. Notably, making the language *harder* is not a dominant driver of boundary crossings. Instead, *simplifying* the phrasing is more often associated with boundary crossings for GPT-4.1-MINI and LLAMA-3.3. One possible explanation is that simplified inputs encourage shorter, more heuristic solution attempts, increasing the chance of committing to a brittle intermediate step; we treat this as a descriptive hypothesis rather than a causal claim.

*Table 4.* Normalized distribution of SBC prediction rates.

| Augmentation | DeepSeek | GPT-4.1 | Llama-3.3 | Phi-4 |
|---|---|---|---|---|
| Add Irrelevant Info | 24.2% | 25.2% | 22.0% | 15.4% |
| Question Reordering | 31.5% | 13.3% | 15.8% | **25.6%** |
| Language Harder | 8.7% | 11.5% | 12.8% | 16.1% |
| Language Simpler | 2.0% | **32.1%** | **30.0%** | 19.0% |
| Subordinate Rev. | **33.6%** | 17.9% | 19.4% | 23.9% |

## 9. Conclusions

We introduced targeted alternation training to improve robustness to meaning-preserving rewrites. For each training question, we run the model multiple times under controlled alternations and merge the resulting intermediate steps into a per-question reasoning graph. We label graph transitions that move from states that can still reach a correct answer (within the constructed graph) to states that cannot, and then learn a boundary-crossing predictor across many questions to surface recurring, model-specific failure transitions. We convert high-confidence predicted crossings into a small set of contrastive rewrite pairs and apply them either via DPO fine-tuning (trainable models) or as in-context demonstrations (closed models).

Across Big-MATH, MMLU-PRO, and DROP, targeted alternation training yields consistent gains over answer-only supervision, random rewrite augmentation, uncertainty-based selection over rewrites, test-time voting over rewrites, and a positive-only ablation, under a leave-one-benchmark-out protocol. In a closed-weight setting, it also improves GPT-5.2 (xhigh) on HLE using in-context demonstrations, showing that targeted rewrite examples can help even without parameter updates. Future work includes extending the approach to settings with tool use and richer feedback signals.

## Impact Statement

This work aims to improve the robustness of large language model reasoning under meaning-preserving question rewrites. By identifying and correcting systematic reasoning failures, it can increase reliability in applications where consistent reasoning is important, such as question answering

and decision support. The method does not introduce or enable harmful uses beyond those already present in existing models.

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

# A. SBC Computation Algorithm

In this appendix section we formalize the SBC computation algorithm mentioned in the main paper here.

---

**Algorithm 1** Compute SBC by Reverse Reachability

---

**Require:** Graph $G_q = (V_q, E_q)$, correct terminals $T_{\text{corr}}$
 1: Build predecessor lists $\text{Pred}[x] = \{p \in V_q : (p, x) \in E_q\}$
 2: $V_{rec} \leftarrow \emptyset$, queue $Q \leftarrow T_{\text{corr}}$
 3: Mark all $t \in T_{\text{corr}}$ as visited; add them to $V_{rec}$
 4: **while** $Q$ not empty **do**
 5:    $x \leftarrow \text{pop}(Q)$
 6:    **for all** $p \in \text{Pred}[x]$ **do**
 7:       **if** $p$ not visited **then**
 8:          mark $p$ visited; add $p$ to $V_{rec}$; push $p$ to $Q$
 9:       **end if**
10:    **end for**
11: **end while**
12: $V_{irr} \leftarrow V_q \setminus V_{rec}$
13: $E_{\text{SBC}} \leftarrow \{(u, v) \in E_q : u \in V_{rec}, \ v \in V_{irr}\}$
14: **Return** $(V_{rec}, V_{irr}, E_{\text{SBC}})$

---

Given a per-question reasoning graph $G_q = (V_q, E_q)$, where each node represents a canonicalized intermediate state and directed edges represent observed one-step transitions between states, we partition nodes into *recoverable* and *irrecoverable* sets with respect to the set of correct terminal nodes $T_{\text{corr}} \subseteq V_q$. A node is *recoverable* if it can reach at least one correct terminal via a directed path in $G_q$; otherwise it is *irrecoverable*. We then define an *SBC edge* as any directed edge $(u, v) \in E_q$ that crosses from a recoverable node to an irrecoverable node. Algorithm 1 computes these sets efficiently via a reverse reachability traversal starting from $T_{\text{corr}}$ (equivalently, a BFS/DFS on the reverse graph), and returns $(V_{\text{rec}}, V_{\text{irr}}, E_{\text{SBC}})$.

# B. SBC Predictor Feature Analysis

This appendix analyzes which *structural properties of reasoning graphs* are most indicative of *Solution Boundary Cuts (SBCs)*. Rather than aiming for a highly expressive predictor, we deliberately use a lightweight, edge-level logistic regression model in order to expose *interpretable and recurring* signatures that distinguish SBC edges from ordinary transitions.

**Training signal and objective.** For each source model, we construct a reasoning graph by repeatedly sampling reasoning trajectories for the same question as we explain in this paper. Each directed edge $(u \rightarrow v)$ is labeled as `SBC` if it separates trajectories that eventually terminate in a *correct* answer from those that terminate in an *incorrect* answer; all other edges are labeled `Non-SBC`. A single logistic regression classifier is trained on a training split, pooling edges across datasets and models.

Importantly, the classifier is not interpreted as producing calibrated probabilities. Instead, its learned coefficients quantify how strongly each feature *discriminates* SBC edges from normal edges in log-odds space.

**Feature interpretation as SBC identifiers.** Each feature $f$ is treated as a continuous or discrete scalar signal defined on edges. The learned coefficient $w_f$ captures how changes in $f$ shift the model's belief that an edge lies on the SBC. A positive weight means that higher values of the feature make an edge *more characteristic* of an SBC; a negative weight means that higher values make it *more characteristic* of a non-SBC edge.

To ground these learned tendencies empirically, we additionally report, for each source model, the mean feature shift

$$\Delta_f = \mu_f(\text{SBC}) - \mu_f(\text{Non-SBC}),$$

which measures how much the feature value differs, on average, between SBC and non-SBC edges. While $\Delta_f$ is not a probability, its sign and magnitude indicate how well the feature *separates* the two edge types in practice.

*Table 5.* Learned logistic regression weights and empirical mean feature shifts ($\Delta$) for identifying SBC edges. Weights quantify how strongly a feature distinguishes SBC edges in log-odds space, while $\Delta$ reflects the observed separation between SBC and non-SBC edges per source model.

| Feature | Weight | Effect on SBC Identification | GPT-5.2 $\Delta$ | DeepSeek $\Delta$ | GPT-4.1-mini $\Delta$ | Llama-3.3-70B $\Delta$ | Phi-4 $\Delta$ |
|---|---|---|---|---|---|---|---|
| is_from_junction | +1.1969 | Strongly increases SBC identifiability | +0.9821 | +0.9792 | +0.9658 | +0.9671 | +0.9750 |
| edge_probability | −0.9854 | High-probability edges are unlikely SBCs | −0.5462 | −0.5493 | −0.5363 | −0.5217 | −0.5375 |
| chain_overlap_ratio | −0.4992 | High redundancy reduces SBC identifiability | −0.3441 | −0.1803 | −0.2222 | −0.2393 | −0.2460 |
| dist_from_junction_src | −2.9310 | SBCs concentrate immediately after junctions | −0.3320 | −0.3921 | −0.3981 | −0.3553 | −0.3661 |
| is_to_merge | −1.1350 | Merging regions suppress SBCs | −0.1599 | −0.2744 | −0.2110 | −0.1884 | −0.2013 |
| dist_from_source_src | +0.6243 | SBC likelihood mildly increases with distance | −0.1427 | −0.1991 | −0.1813 | −0.1295 | −0.1483 |
| depth_normalized | −0.0560 | Weak late-stage suppression of SBCs | −0.1351 | −0.1671 | −0.1572 | −0.1115 | −0.1253 |

**Updated feature summary.** Table 5 reports the learned logistic regression weight for each feature, along with the per-model empirical mean shifts. Together, these quantities describe both the *directional influence* learned by the model and the *observed discriminative signal* present in the data.

**Structural interpretation.** Several consistent patterns emerge. Edges originating from junctions are by far the strongest identifiers of SBCs, indicating that failures are typically decided at points of genuine choice. Conversely, edges that are dominant (high transition probability), redundant across chains, or located downstream from reconverging structure are strongly associated with non-SBC behavior. Distance-based features play a secondary role, suggesting that *local structural context* is more informative than absolute position in the reasoning chain.

**Summary.** Overall, these features do not merely correlate with SBCs individually; together they describe a coherent failure signature: SBCs arise at low-probability, non-redundant transitions immediately following branching points, before trajectories have a chance to reconverge. This explains why even a simple linear model can reliably identify recurring solution boundaries, while remaining insensitive to idiosyncratic or one-off errors.

# C. Augmentation Prompts

**Shared component.** All augmentation prompts in this work share the same base instruction. This component enforces semantic preservation, option consistency, and self-contained rewriting, while delegating the actual transformation to the augmentation-specific rules below.

```
# System Prompt

- You only rewrite the user's content.
- Do not answer the question or provide any solution.
- Rewrite the entire user input into a self-contained input without omitting any
↪   information.
- Preserve the original meaning and the correct answer.
- Keep all answer options (if any) present and in the same order.
- The output must be a full rewritten version of the original input and must not assume
↪   any prior context.
```

## C.1. Question Reordering

**Augmentation description.** *Question Reordering* changes the surface order of question components while preserving semantics and the answer.

```
# Required Augmentation (extends Shared Base Prompt)

- Reorder the components of the question while maintaining its original meaning and
↪   answer.

# Example
```

```
Original:
"There are 3 pufferfish in the market.
How many pufferfish are in the market?"

Rewritten:
"How many pufferfish are in the market, given that there are 3 pufferfish available for
↪   sale?"
```

### C.2. Subordinate Reversing

**Augmentation description.** *Subordinate Reversing* moves subordinate clauses to different positions while keeping identical semantics.

```
# Required Augmentation (extends Shared Base Prompt)

- Rewrite the question by moving subordinate clauses to different positions while
↪   preserving meaning.

# Example

Original:
"There are 3 pufferfish in the market because the vendor restocked this morning.
How many pufferfish are in the market?"

Rewritten:
"Because the vendor restocked this morning, there are 3 pufferfish in the market.
How many pufferfish are in the market?"
```

### C.3. Language Simpler

**Augmentation description.** *Language Simpler* rewrites the input using simpler, more accessible language while toggle preserving meaning and answer.

```
# Required Augmentation (extends Shared Base Prompt)

- Rewrite the question using simpler vocabulary and clearer phrasing while preserving
↪   meaning.

# Example

Original:
"There are 3 pufferfish in the market.
How many pufferfish are in the market?"

Rewritten:
"There are 3 pufferfish at the market.
How many pufferfish are there?"
```

### C.4. Language Harder

**Augmentation description.** *Language Harder* rewrites the question using more complex or formal phrasing. If a modified word appears in the answer options, it must be updated consistently.

```
# Required Augmentation (extends Shared Base Prompt)

- Rewrite the question using more complex and formal language while preserving meaning.

# Example
```

```
Original:
"There are 3 pufferfish in the market.
How many pufferfish are in the market?"

Rewritten:
"Within the confines of the marketplace, there exist three individual specimens of
↪   pufferfish.
What is the total number of pufferfish present in this market environment?"
```

### C.5. Add Irrelevant Information

**Augmentation description.** *Add Irrelevant Information* injects unrelated or esoteric details that do not affect the solution path.

```
# Required Augmentation (extends Shared Base Prompt)

- Add unrelated or descriptive details that increase length but do not affect the
↪   answer.

# Example

Original:
"There are 3 pufferfish in the market.
How many pufferfish are in the market?"

Rewritten:
"In a seaside market filled with the sound of waves and colorful coral decorations,
there are 3 pufferfish available for sale.
How many pufferfish are in the market?"
```

## D. Human Equivalence Spotchecks for Rewrite Filtering

To ensure that automatically filtered rewrites are genuinely meaning-preserving, we conducted random human spotchecks on the outputs of the semantic equivalence filter used during solution-trace sampling (Section 3.1). The purpose of this audit is not to estimate an exact error rate, but to verify that the filter is conservative and does not admit systematic semantic drift.

We randomly sampled a small subset of rewrite–original pairs accepted by the equivalence judge and asked human annotators to assess whether each rewrite preserves the original question's intent and correct answer. Annotators were instructed to judge semantic equivalence only, ignoring stylistic differences, verbosity, or surface reordering. Disagreements were resolved by majority vote.

The spotchecks indicate that the vast majority of accepted rewrites ($\approx 98\%$) are indeed meaning-preserving, with observed failures being rare and non-systematic (e.g., subtle scope narrowing or accidental strengthening of constraints). Importantly, we did not observe clusters of semantically invalid rewrites tied to any specific rewrite family.

These results support the use of the automated equivalence filter as a high-precision gate in our pipeline. Given that rewrites are used solely to elicit diverse solution traces—not to select answers or define supervision—the residual noise observed in these spotchecks is unlikely to materially affect the structure of the induced reasoning graphs or the downstream identification of solution-boundary crossings.

## E. Discretizing and Annotating Solution Traces

This appendix details the procedure used to convert free-form solution traces into a canonical, step-level representation and to attach lightweight semantic annotations. These operations are applied uniformly across datasets and models and serve only to support structural alignment and reasoning-graph construction. We first discuss how step discretization interacts with the localization of solution-boundary crossings (SBCs), and then provide the exact prompts used for step segmentation and semantic annotation.

### E.1. Step Discretization Effect on SBC location

Step discretization is a pragmatic approximation: continuous free-form reasoning traces are segmented into discrete units to enable alignment, merging, and graph construction. This segmentation may alter the apparent granularity of reasoning transitions and, in turn, the exact placement of inferred boundaries in the aggregated reasoning graph.

While merging errors may shift the location of boundaries in the global graph, the SBC construction guarantees that for any given failing trace, a boundary-crossing edge exists that does not conflict with a successful continuation from the same prefix.

In this sense, discretization affects the *representation* of where a boundary is observed but not the *existence* of a solution-boundary crossing itself. SBCs are therefore robust to reasonable variations in step segmentation, provided that prefix consistency is preserved across compared traces.

### E.2. Step Discretization Prompt

You are given a question and a solution trace written as free-form text.

Your task is to insert explicit step delimiters into the solution trace. Each step should correspond to a single intermediate reasoning action, decision, or update.

Rules: - Preserve the original wording exactly. - Do not add, remove, rephrase, or correct any content. - Only insert step delimiters. - Do not explain or justify the steps. - The final answer should also be marked as a step.

Output the trace with clear step boundaries.

### E.3. Semantic Annotation Prompt

You are given: (1) the original question. (2) a discretized solution trace consisting of ordered steps. (3) the final answer produced by the model.

For each step, assign: 1. A slot or variable tag describing the main quantity, entity, or concept the step refers to. 2. An operation label chosen from the following set: - RETRIEVE: restating or extracting information from the question or context - INFER: deriving new information from previous steps (e.g., arithmetic, logical inference) - GENERATED: introducing background knowledge not explicitly stated - OTHER: steps that do not fit the above categories

Rules: - Do not change the text of the steps. - Do not judge correctness. - Do not add new reasoning. - Use coarse, high-level labels; precision is not required.

Output the annotated steps in order.

### E.4. Notes on Usage

The outputs of these prompts are used solely to support merging and alignment of steps across sampled traces when constructing per-question reasoning graphs. They do not affect answer selection, correctness evaluation, or supervision of the target model. Residual annotation noise primarily influences step granularity and merge behavior rather than the identification of solution-boundary crossings.

## F. Edge Features for the Boundary-Crossing Predictor

This appendix describes the edge-level features used by the boundary-crossing predictor (Section 4.2). These features are designed to capture simple, interpretable properties of local graph structure and empirical execution behavior. They are not intended to fully characterize the semantics of reasoning steps, but rather to summarize where a transition occurs in the reasoning graph and how it is used by sampled solution traces.

**Structural features.** We include indicators of local graph connectivity around a directed edge $(u \to v)$. These capture whether the transition originates from or leads into a structurally ambiguous region of the graph. Concretely, we use whether $u$ is a branching node (out-degree $> 1$), whether $v$ is a merge node (in-degree $> 1$), as well as the out-degree of $u$ and the in-degree of $v$. These features reflect the intuition that irrecoverable failures often arise at decision points or enter regions

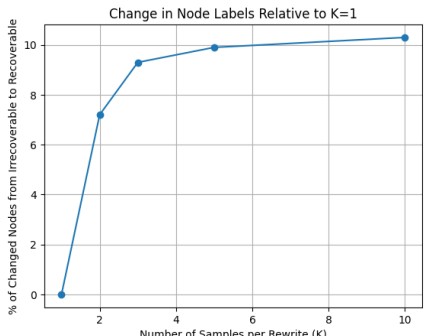

*Figure 3.* Change in recoverability labels as the number of sampled traces per rewrite increases. The plot reports the percentage of nodes labeled irrecoverable at $K = 1$ that become recoverable under larger sampling budgets.

where multiple trajectories collapse.

**Positional features.** To describe where an edge lies within a typical reasoning trajectory, we compute normalized distances in the graph. These include the shortest-path distance from a source to $u$ and $v$, the shortest-path distance from $u$ and $v$ to any terminal node, and the distance from the nearest upstream branching node. All distances are normalized to $[0, 1]$ within each per-question graph, so values are comparable across questions of different lengths.

**Probability and usage features.** We capture how frequently a transition is taken under stochastic decoding by including the empirical transition probability of $(u \rightarrow v)$, estimated from the sampled traces, as well as the marginal visitation probability of the source node $u$.

**Redundancy and overlap features.** Finally, we include a chain-overlap ratio (or more intuitively edge probability) that measures how much the sets of traces going into $u$ continue to $v$. Intuitively, low overlap indicates divergence into more isolated trajectory regions, while high overlap reflects shared or dominant reasoning paths.

## G. Reasoning Graph Further Ablations

### G.1. Sensitivity to the Number of Sampled Traces

Our main experiments use $K = 2$ sampled solution traces per retained rewrite and original question. Since recoverability and irrecoverability are defined with respect to the constructed reasoning graph, this choice affects the observed graph topology: additional samples may reveal new continuations from a node that was previously labeled irrecoverable. Therefore, irrecoverability should be interpreted as an observational property of the sampled graph, not as a claim that recovery is impossible in principle.

To evaluate sensitivity to the sampling budget, we vary the number of sampled traces per rewrite,

$$K \in \{1, 2, 3, 5, 10\},$$

using GPT-4.1-mini, while keeping the rewrite-generation and graph-construction pipeline fixed. For each value of $K$, we reconstruct the per-question reasoning graph, recompute the recoverable and irrecoverable node sets, and compare the resulting node labels to the labels obtained with $K = 1$.

We report the percentage of nodes that change from irrecoverable under $K = 1$ to recoverable under a larger sampling budget:

$$C(K) = 100 \cdot \frac{\sum_q \left| \left\{ v \in V_q^{(1)} \cap V_q^{(K)} : v \in V_{\text{irr}}^{(1)} \wedge v \in V_{\text{rec}}^{(K)} \right\} \right|}{\sum_q \left| V_{\text{irr}}^{(1)} \cap V_q^{(K)} \right|}.$$

This measures how often additional sampling uncovers a path from a previously irrecoverable node to a correct terminal.

As shown in Figure 3, most of the change occurs when moving from one sampled trace to two sampled traces per rewrite. Increasing the budget further continues to reveal additional recoverable continuations, but the curve flattens after $K = 2$–$3$,

indicating diminishing returns from higher sampling budgets. We therefore use $K = 2$ in the main experiments as a practical trade-off between computational cost and recoverability-estimation quality.

This ablation also clarifies the interpretation of the SBC. Higher $K$ can change some node labels by exposing recovery paths that were not sampled at lower $K$, so the SBC should be understood as the boundary induced by the observed trace distribution. Exhaustive high-$K$ sampling may further refine rare recovery paths, but the observed trend suggests that the dominant correction occurs already at low sampling budgets.

### G.2. Ablation on Node-Merging Features

In Section 3, node merging uses three sources of information: the step text embedding, the slot/variable tag embedding, and a coarse operation label. The slot/variable tag is intended to prevent merging steps that use semantically similar wording but refer to different quantities or entities, while the operation label is intended to prevent merging steps that perform different reasoning operations despite surface similarity.

To evaluate whether these additional annotations improve merge quality, we perform a feature ablation over the node-merging rule. Specifically, we compare four variants: using both slot/variable tags and operation labels, removing only the slot/variable tag, removing only the operation label, and removing both. For each variant, we sample candidate merges and evaluate them with an LLM-as-a-judge audit using GPT-5. The judge is given the original question, the reasoning chains containing the candidate steps, and the resulting merged node. It then returns a binary decision indicating whether the merge is semantically valid.

*Table 6.* Node-merging ablation. Values report the fraction of candidate merges judged semantically valid by GPT-5. Higher values indicate better merge precision.

|                  | Without Variable Tag | With Variable Tag |
| ---------------- | -------------------- | ----------------- |
| Without Op. Label | 0.87                 | 0.91              |
| With Op. Label    | 0.92                 | **0.94**          |

Table 6 shows that slot/variable tags and operation labels consistently improve merge precision: adding them increases the judged validity of candidate nodes being semantically equivalent.

### G.3. Effect of Boundary-Crossing Predictor Quality

The boundary-crossing predictor is used to prioritize candidate SBC edges for constructing targeted alternation pairs. Therefore, predictor quality may affect downstream performance: a more accurate predictor should select cleaner recoverable-to-irrecoverable transitions, while a weaker predictor may introduce noisier alternation pairs.

To test this sensitivity, we replace the GNN predictor used in the main experiments with the next-best predictor from our architecture ablation, a Graph Transformer. All other components are kept fixed: we reconstruct targeted alternation pairs using the alternative predictor, retrain GPT-4.1-mini with the same training protocol, and evaluate on the two datasets where the baseline model has the lowest initial performance, MMLU-PRO and Big-MATH.

Table 7 shows that replacing the GNN with the Graph Transformer leads to a modest downstream degradation, but does not eliminate the gains of the method. This suggests that predictor quality affects the strength of the selected alternation pairs, yet the overall pipeline is not brittle to a single predictor architecture. A stronger boundary-crossing predictor yields cleaner training pairs and better downstream performance, while a slightly weaker predictor still preserves the main effect. We observe the same qualitative trend for the other evaluated target models.

### G.4. Sensitivity to the Annotation Model

To assess the sensitivity of our method to the LLM used for discretization and annotation, we rerun the full pipeline with two different annotators: GPT-5-mini, which is the annotator used in the main experiments, and GPT-4o-mini. We report results for GPT-4.1-mini fine-tuned with our method on MMLU and DROP in Table 8.

The results are similar across annotators, indicating that the pipeline is not highly sensitive to the particular LLM used for discretization and annotation. In this setting, replacing GPT-5-mini with GPT-4o-mini slightly reduces downstream performance. This suggests that the gains are not driven by a single strong annotator. We observed the same overall trend for

*Table 7.* Effect of the boundary-crossing predictor on downstream accuracy. We compare the main GNN predictor AUC against the next-best Graph Transformer predictor while keeping the rest of the pipeline fixed.

| SBC predictor | MMLU-PRO | Big-MATH |
|---|---|---|
| GNN (main) | 78.0 | 74.8 |
| Graph Transformer | 77.5 | 73.1 |

*Table 8.* Sensitivity of our method to the LLM used for discretization and annotation. Results are reported for GPT-4.1-mini fine-tuned with our method.

| Annotation Model | MMLU | DROP |
|---|---|---|
| GPT-4o-mini | 77.0 | 89.0 |
| GPT-5-mini | 78.1 | 91.0 |

the other evaluated models as well.

# H. Additional Results

## H.1. Evaluation on Transformed Test Questions

The main experiments evaluate models on the original, unaugmented test questions. To further verify that the learned robustness transfers to DLR questions, we also evaluate the trained GPT-4.1-mini on a transformed version of the test set. For each test question, we generate meaning-preserving transformations using the same rewrite templates and equivalence filter used in our training pipeline. No test question is used for graph construction, pair selection, training, or hyperparameter selection.

| Model | Original Test | Transformed Test |
|---|---|---|
| GPT-4.1-mini | 76.5 | 72.0 |
| GPT-4.1-mini + Targeted Alternation | 81.5 | 76.1 |

*Table 9.* Average accuracy across MMLU-PRO, Big-MATH, and DROP on the original test questions and on DLR test questions.

Table 9 reports the average accuracy across MMLU-PRO, Big-MATH, and DROP. The base model drops from 76.5 on the original test set to 72.0 on the transformed test set, confirming that meaning-preserving transformations induce a meaningful robustness challenge. After targeted alternation training, performance improves on both the original and transformed test sets, reaching 81.5 and 76.1, respectively. Thus, the gains are not limited to the original benchmark phrasing; they also persist under semantically equivalent transformed inputs.

## H.2. SBC Predictor Failure Analysis

We analyze misclassified edges of the boundary-crossing predictor on held-out questions. Figures 4 and 5 show representative false-positive and false-negative cases. These examples are consistent with the structural patterns observed in our feature analysis, where junction-related features and edge transition probability are among the strongest predictors of SBC edges.

**False positive: symmetric fork.** A common false positive occurs at a binary junction where one outgoing edge is the true SBC edge, while the sibling edge leads to a correct continuation. Since the two sibling edges share nearly identical local structural features—the same source node, depth, out-degree, and similar transition probability—the predictor assigns high SBC likelihood to both. As a result, the correct edge is incorrectly flagged as boundary-crossing. This failure mode reflects the predictor's reliance on junction-local structure: edges leaving branching nodes are often predictive of SBCs, but in symmetric forks, local features can be insufficient to distinguish the true cut from its correct sibling.

**False negative: high-probability incorrect path.** A representative false negative occurs when the true SBC edge is also the dominant outgoing transition from a junction. In our feature analysis, edge transition probability is generally a negative predictor of SBC labels, since high-probability edges often correspond to stable continuations. This heuristic is useful in

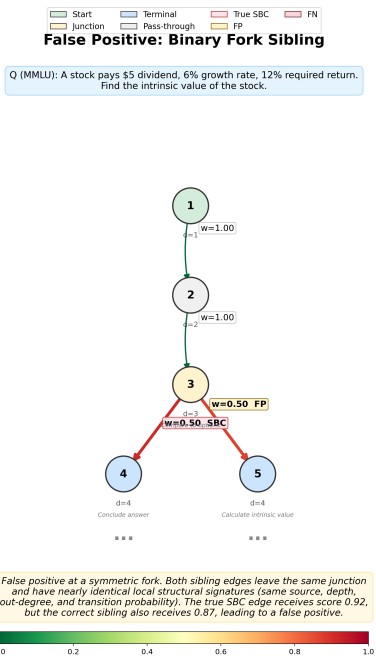

*Figure 4.* Representative false-positive SBC prediction. The predictor incorrectly flags a correct sibling edge as boundary-crossing because it shares local structural features with the true SBC edge.

aggregate, but fails when the model's most common next step is already irrecoverable. In such cases, the predictor assigns low SBC confidence to the true boundary-crossing edge and misses the onset of the error.

Overall, these examples show that the predictor captures useful structural regularities, but can fail when local graph features are not sufficient to disambiguate sibling continuations or when the statistically dominant trajectory is itself incorrect.

## I. Reasoning-Graph Construction Details

This appendix provides additional implementation details and design choices for the construction of per-question reasoning graphs. We focus on (i) how sampled solution traces are merged into a single empirical graph, and (ii) how hyperparameters governing graph construction are selected and validated. The procedures described here are fixed across all experiments unless stated otherwise.

### I.1. Graph Construction Procedure

For each base question $q$, we construct a directed reasoning graph $G_q = (V_q, E_q)$ from a collection of sampled solution traces obtained under stochastic decoding and meaning-preserving rewrites. The graph is intended as an empirical summary of the model's observed intermediate reasoning states and transitions, not as a complete or faithful model of its internal computation.

**Nodes.** Each node represents a canonicalized intermediate step, as defined in Section 3.2. A step is characterized by three components: (i) the step text span, (ii) a slot/variable tag describing the main quantity or concept involved, and (iii) a coarse operation type (e.g., RETRIEVE, INFER). We embed the step text and slot tag using a fixed embedding model and retain the operation type as a discrete attribute.

During graph construction, a newly observed step is mapped to an existing node if and only if it matches an existing node under a deterministic merge rule: the operation types must be identical, and the weighted cosine similarity between the concatenated embeddings must exceed a predefined threshold. If no existing node satisfies this criterion, a new node is created. This procedure ensures that only semantically similar and operationally equivalent steps are merged.

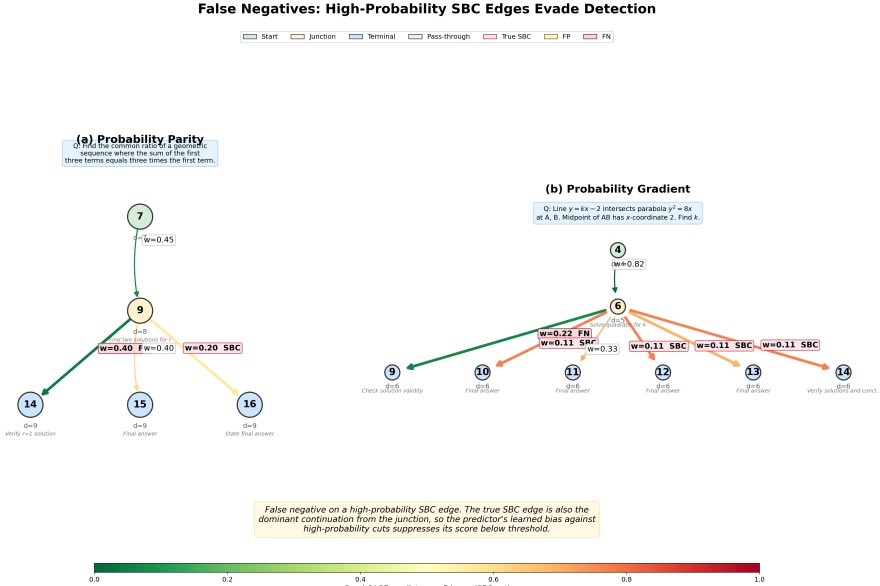

*Figure 5.* Representative false-negative SBC prediction. The predictor misses the true boundary-crossing edge because the incorrect transition is also the dominant outgoing path.

**Edges.** Directed edges correspond to observed step-to-step transitions within individual traces. For each trace, consecutive mapped nodes $(u, v)$ induce a directed edge. Edge multiplicity is retained by storing a count of how many times each transition is observed across all traces for the question. These counts are later normalized when computing empirical transition probabilities or edge-level features, but the raw counts are preserved during graph construction.

**Graph structure.** Because steps can recur across different traces—and, in rare cases, within a single trace—the resulting graph may contain cycles. We therefore treat $G_q$ as a general directed graph rather than a tree or DAG. No acyclicity or ordering constraints are imposed beyond the observed temporal order within traces although the case is not real, but it was observed is less than $1\%$ of the graphs and does not have an immediate effect on the proposed model.

**Efficiency considerations.** To avoid quadratic comparisons when matching new steps to existing nodes, we index node embeddings using approximate nearest-neighbor search. Candidate matches are retrieved from this index and then filtered by the exact merge rule described above. This optimization affects only efficiency, not the semantics of the constructed graph.

**Interpretation.** The reasoning graph is an empirical object: it reflects which intermediate states and transitions the model actually visits under the sampled conditions. Absence of an edge or path should not be interpreted as impossibility, only as lack of observation under the given sampling budget. All downstream analyses, including Solution Boundary Cut (SBC) computation, are defined strictly with respect to the constructed graph.

### I.2. Hyperparameter Selection and Validation

Graph construction involves a small number of hyperparameters that control step merging and edge statistics. These hyperparameters are selected in a model-agnostic and dataset-agnostic manner using a held-out validation split, and are fixed before any evaluation or training experiments.

**Tuned parameters.** The primary hyperparameters include: (i) the similarity threshold used to merge steps into a single node, (ii) the relative weighting between step-text embeddings and slot/variable-tag embeddings in the similarity computation. No hyperparameters depend on correctness labels or SBC annotations.

**Tuning protocol.** Hyperparameters are tuned on a validation subset disjoint from all reported training and test splits. The objective of tuning is structural stability rather than downstream accuracy: we select parameters that produce graphs with (a) reasonable node reuse across traces, (b) non-trivial branching structure, and (c) stable graph statistics under repeated

resampling of traces. In practice, this corresponds to avoiding degenerate regimes where graphs collapse to near-linear chains (over-merging) or fragment into many singleton nodes (under-merging).

**Statistical robustness.**  To assess robustness, we resample solution traces for the same questions under different random seeds and verify that aggregate graph statistics—such as node count, edge count, and degree distributions—vary smoothly rather than discontinuously. Hyperparameter settings that produce highly unstable graphs under small resampling perturbations are rejected.

**Isolation from evaluation.**  Crucially, hyperparameter tuning is performed without access to gold answers, SBC labels, or downstream performance metrics. This ensures that the reasoning graphs themselves do not encode task-specific supervision and that subsequent SBC computation remains a purely structural analysis conditioned only on observed trajectories.

**Limitations.**  The chosen hyperparameters reflect a trade-off between granularity and robustness under finite sampling. While different settings may yield alternative graph resolutions, our ablations indicate that the qualitative SBC patterns described in the paper are stable across a broad range of reasonable values. We therefore treat these hyperparameters as implementation details rather than model-specific optimizations.

## J. Computational Cost of Offline Data Construction

Our method incurs its main overhead during a one-time offline data-construction stage. This cost comes from three components: (i) generating meaning-preserving rewrites for each training question, (ii) sampling multiple solution traces from the target model for the original question and its rewrites, and (iii) discretizing and annotating those traces in preparation for graph construction and clustering.

Let $R$ denote the number of accepted rewrites per question, $K$ the number of sampled traces per input, $|q|$ the average question length in tokens, and $|\tau|$ the average length of a sampled solution trace. The total token cost per training question can be decomposed as

$$\mathcal{C}_{\text{total}} = \mathcal{C}_{\text{rewrite}} + \mathcal{C}_{\text{trace}} + \mathcal{C}_{\text{annot}},$$

where

$$\mathcal{C}_{\text{rewrite}} = O(R|q|),$$

$$\mathcal{C}_{\text{trace}} = O((R+1)K(|q| + |\tau|)),$$

and

$$\mathcal{C}_{\text{annot}} = O((R+1)K|\tau|).$$

In our main setup, we use $R = 5$ rewrites and $K = 2$ sampled traces per input, so each training question produces traces for the original question plus five rewrites, for a total of $(R+1)K = 12$ sampled traces. Under the mild assumption that $|q| = O(|\tau|)$, the overall cost remains linear in the total number of sampled traces and is dominated by trace generation and annotation.

Importantly, this overhead is incurred only once during offline data construction. For trainable models, deployment-time inference uses the resulting fine-tuned model directly and therefore does not require repeating the rewrite-generation or graph-construction pipeline at test time.

Figure 6 shows that targeted alternation training consistently improves over the baseline for all four evaluated models. Although the method introduces additional offline token cost, the resulting gains are positive across all backbones. The figure also shows that the cost–benefit trade-off is model-dependent: some models obtain larger improvements for a relatively moderate increase in average token usage, while others require higher token budgets for somewhat smaller gains. Overall, the results indicate that the additional annotation and graph-construction cost is associated with consistent downstream benefits rather than isolated gains on a single model.

As a concrete example, DeepSeek-V2-Lite exhibits the largest relative improvement in the figure, while GPT-4.1-mini, LLaMA-3.3-70B, and Phi-4 also show clear positive gains. We emphasize that this analysis is intended to quantify the overhead of the method, not to claim that all models have the same efficiency profile. Rather, it shows that the proposed pipeline yields a favorable accuracy–cost trade-off across diverse model families.

*Figure 6.* Average trade-off between token cost and downstream gains, aggregated over Big-MATH, MMLU-PRO, and DROP. Each open-circle marker shows the baseline model, and each filled marker shows the same model after targeted alternation training. The horizontal axis reports the average tokens per answer, and the vertical axis reports relative improvement over the corresponding baseline.

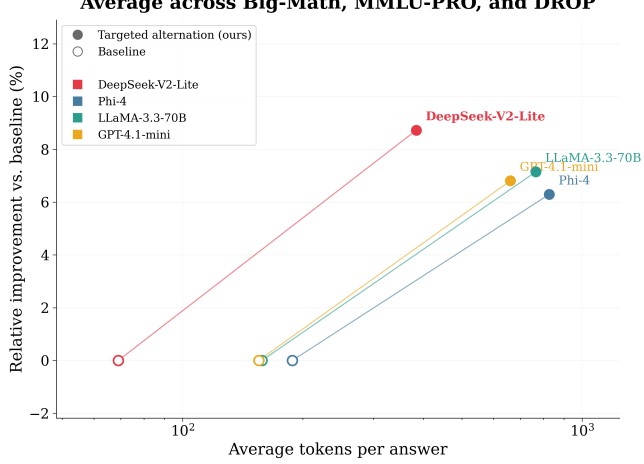

