# OpenReview forum: "Learning Rewrite-Invariant Reasoning with Targeted Alternation Training"
_ICML.cc/2026/Conference — ICML 2026 regular_

### Official Review · Reviewer_NwB3 · 2026-02-26

**Soundness:** 4
**Presentation:** 3
**Significance:** 3
**Originality:** 3
**Overall Recommendation:** 4
**Confidence:** 4

**Summary:**

This paper proposed an end-to-end procedure that perturbs questions with rewrites, samples multiple solution traces, and aggregates them into a reasoning graph of recurring intermediate steps. Furthermore, this paper identified the earliest divergence after which a trace can no longer reach a correct answer as the Solution Boundary Cut. Extensive experiments on MMLU-Pro, Big-MATH, and DROP demonstrated the effectiveness of the proposed method.

**Compliance With Llm Reviewing Policy:**

Affirmed.

**Final Justification:**

I have no further questions. I maintain my positive score.

**Key Questions For Authors:**

1. The author should provide a case study of boundary-crossing predictors, especially examples of prediction errors.
2. The proposed method appears to be somewhat similar to step-dpo[1], and the authors should discuss these related works.

[1] Step-DPO: Step-wise Preference Optimization for Long-chain Reasoning of LLMs

**Limitations:**

yes

**Strengths And Weaknesses:**

Strengths
1. The paper is clear and well structured.
2. The proposed method has high generalization ability and can theoretically be extended to most language tasks.
3. The paper constructs extensive experiments to demonstrate the effectiveness of the proposed method.

Weaknesses
1. The framework is quite complex, and the author did not provide the source code, so its reproducibility is relatively low.
2. The experiment had a small scope, only including a set of math-related tests.
3. Some details are missing, such as the statistics of the training set, the structure of embedding model E, and the boundary-crossing predictor.

---

> ### Author Rebuttal · Authors · 2026-03-31
>
> **1.** Code and data are already publicly available (see line 109: “Code and data available here”), where clicking on “here” directs to the full data and code repository.
>
> **2.** We evaluate on three challenging, state-of-the-art datasets spanning multiple domains beyond math (please see all datasets we detail in Section 5.2): MMLU-PRO (broad multi-domain reasoning), Big-MATH (multi-step symbolic/numeric computation), and DROP (discrete reasoning over passages). A total of 15K questions in total.
> This selection is intended to cover complementary reasoning regimes rather than a single domain. We will add in the discussion that expanding to additional datasets is a natural direction as resources allow.
>
> **3.**
>
> *Training set statistics:* We use the official training splits provided by the original benchmark authors (MMLU-PRO, Big-MATH, and DROP). For each training question, we generate five meaning-preserving rewrites and sample $K=2$ reasoning trajectories per rewrite. The final SBC training dataset contains approximately 60K questions paired with structured reasoning chains, totaling 17.2M tokens. The data is distributed roughly equally across the three benchmarks.
>
> *Embedding model:* We use BGE (specifically bge-large-en-v1.5 with the size 335M parameters) as the embedding model E referenced in Section 3.2. We will name it explicitly and cite it in the revision .
>
> *Boundary-crossing predictor:* The solution-boundary predictor used in our main experiments (Table 1) is the GNN model described in Sec. 7.2, with its comparison to alternative architectures reported in the ablation study (Table 3). Specifically, it uses a GraphSAGE encoder with residual connections and LayerNorm, followed by an edge-prediction MLP that takes the concatenation of source and destination node embeddings together with raw edge features, and a min-cut decoder applied at inference time. We will add an explicit reference to this model in Sec. 4.2 and provide full hyperparameters and implementation details in the appendix.
>
> **4.** We analyze misclassified edges of the boundary-crossing predictor on held-out questions, and include one representative false positive and one false negative case study. These examples are consistent with the structural biases identified in our feature analysis, where junction-related features and edge probability are among the strongest predictors of SBC edges.
>
> *False positive: symmetric fork.*
> A common false positive occurs at a binary junction where one outgoing edge is the true SBC edge and the sibling edge leads to a correct continuation. Because the two sibling edges share nearly identical local structural features (same source node, same depth, same out-degree, and similar transition probability), the predictor assigns both high SBC likelihood, incorrectly flagging the correct edge as boundary-crossing. This failure mode is consistent with the predictor’s learned reliance on junction-local structure: edges leaving branching nodes are often predictive of SBCs overall, but in a symmetric fork the local features may be insufficient to distinguish the true cut from its correct sibling. Please see an illustrative example by clicking [HERE](https://drive.google.com/file/d/1fkneH4qqtNlItjFaTxRQU_iSbk0Oc-cr/view?usp=sharing).
>
> *False negative: high-probability incorrect path.*
> A representative false negative occurs when the truly boundary-crossing edge is also the dominant outgoing transition from a junction. Since edge probability is a strong negative predictor of SBCs in our analysis, the model has learned that high-probability edges are usually not cuts. This heuristic is statistically useful overall, but it fails in cases where the model’s most common next step is already the irrecoverable one. In such examples, the predictor assigns low SBC confidence to a genuinely boundary-crossing edge and misses the error onset.
> We will add these two examples as local graph visualizations in the appendix, showing the question, the relevant subgraph, the ground-truth SBC label, and the predictor’s score, to make both the predictor’s strengths and its limitations more transparent. Please see illustrative graphs by clicking [Here](https://drive.google.com/file/d/1qG_kC7Kz1PDfbDzTFBbKCn5PxZ3I7Wu0/view?usp=sharing).
>
>
> **5.** Step-DPO provides local step-level supervision by preferring a better next step over a worse one. In contrast, SBC identifies the transition from recoverable to unrecoverable reasoning and trains on those boundary cases. Thus, Step-DPO improves step quality, while SBC targets robustness to reasoning collapse. We will add this to the related work.

---

> > ### Author Rebuttal · Reviewer_NwB3 · 2026-04-03
> >
> > I have no further questions, and I'm happy to maintain my positive score.

---

### Official Review · Reviewer_2daF · 2026-03-10

**Soundness:** 2
**Presentation:** 3
**Significance:** 2
**Originality:** 2
**Overall Recommendation:** 4
**Confidence:** 3

**Summary:**

This paper proposes the Targeted Alternation Training (TAT) framework, which applies semantically-preserving rewrites to each training problem (paraphrases, format transformations, benign distractors, etc.), samples multiple reasoning trajectories from the target model, and aggregates these trajectories into a Reasoning Graph. Nodes in the graph represent recurring intermediate reasoning steps, while edges represent transitions between steps. This design transforms the model's reasoning process from a black box into something analyzable. The method's effectiveness is validated through cross-dataset generalization experiments on MMLU-Pro, Big-MATH, and DROP, as well as evaluations of GPT-5.2 on Humanity's Last Exam (HLE).

**Compliance With Llm Reviewing Policy:**

Affirmed.

**Final Justification:**

The rebuttal meaningfully addressed several of my concerns.

**Key Questions For Authors:**

See Weaknesses

**Limitations:**

yes

**Strengths And Weaknesses:**

Strengths

1. A graph method is proposed to precisely localize the boundary at which a reasoning trajectory transitions from a "recoverable" state to an "irrecoverable" one. This means failure is no longer a coarse-grained label but can be pinpointed to a specific reasoning step.
2. The high-confidence failure boundaries identified by SBC are used to automatically construct "alternation pairs", which are then used to train the model via DPO fine-tuning or in-context learning.
3. Consistent improvements are achieved across three benchmarks with cross-dataset generalization also validated.

Weaknesses

1. Unrecoverability is defined within the constructed graph, and only K=2 trajectories are sampled per rewrite. A node being labeled "unrecoverable" may simply reflect the fact that no correct trajectory happened to be sampled from that node, rather than a genuine inability of the model to recover from it. As the number of samples increases, nodes previously deemed "unrecoverable" may become recoverable. This means SBC is an approximation heavily affected by sampling noise, yet the paper provides insufficient discussion of this issue.
2. In Table 1, the accuracy of DeepSeek-V2-Lite on MMLU-PRO drops from 60.1% to 53.8% after standard supervised fine-tuning—an anomalous degradation that is left entirely unexplained, undermining confidence in the reliability of the experimental implementation.
3. The computational cost of the entire pipeline is never reported, making it impossible to assess the practical overhead of the method relative to the baselines. In terms of absolute gains, most results improve by 1–7 percentage points, and for some stronger model/dataset combinations (e.g., LLaMA-3.3 on MMLU-PRO), the gains are quite modest. The paper provides no cost-benefit analysis to justify whether the improvements warrant introducing such a complex pipeline for real-world deployment.
4. Table 3 shows predictor AUC values in the range of 70–83, implying that a non-trivial proportion of boundary-crossing edges are misclassified. While the paper acknowledges that "perfect classification of every failure edge is not required," it does not analyze how much the predictor's precision affects the quality of the resulting training samples and downstream accuracy. This critical sensitivity analysis is absent.
5. The statistical significance of the HLE results is questionable, and Figure 1 is potentially misleading. The accuracy improves from 35.4% to 38.1%, an absolute gain of only approximately 2.7 percentage points. The paper claims a 95% confidence interval, but the error bars in Figure 1 appear quite wide. The paper neither explicitly reports whether this difference is statistically significant under a formal hypothesis test, nor states the sample size of the HLE evaluation set.

---

> ### Author Rebuttal · Authors · 2026-03-31
>
> Thank you for your detailed review, please find below the clarifications to your questions:
>
> **1. Unrecoverability could be a result of under-sampled graphs:**
>
> **Response** Unrecoverability is defined with respect to the constructed graph and may be affected by sampling, as discussed in Sec. 4.1. To evaluate this sensitivity, we include here an ablation varying the number of sampled trajectories per rewrite (K=1, 2, 3, 5, 10) using GPT-4.1-mini, resulting in over 50 traces per question (5 question augmentations x K). See the output graph in this [link](https://drive.google.com/file/d/12YycbpJlwL66fQlbBTB6f2iKmZJi57hL/view?usp=sharing).
> In practice, we select K=2 as a sweet spot between computational efficiency and boundary estimation accuracy. More exhaustive sampling is unlikely to materially change the learned boundary, and exploring higher-K regimes at scale remains future work. We will include this analysis in the appendix and clarify this point in the final version. We observe a similar trend for other models as well.
>
> **2. Accuracy drop**
>
> **Response** We do not believe it indicates an implementation issue, since the same SFT and evaluation pipeline was applied across all backbones, while the degradation is concentrated in DeepSeek-V2-Lite rather than appearing uniformly. A plausible contributing factor to the performance drop is that DeepSeek-V2-Lite is an MoE model with only 2.4B active parameters per token (despite 15.7B total parameters), which may make it more sensitive to **final answer-only SFT** than the other backbones we tested. Prior work [1][2][3][4] shows that SFT can induce backbone-dependent ability interference and catastrophic forgetting in DeepSeek small models, so this behavior is consistent with model-specific negative transfer rather than a systematic experimental artifact. We will include a short discussion on this in the paper.
>
> Relevant prior work:
>  [1] Fu et al., Findings of ACL 2024 — Instruction tuning can degrade MMLU due to alignment tax (overfitting to instruction distribution).
> [2] Gupta et al., Findings of NAACL 2025 — Standard SFT causes performance drops via overfitting and forgetting.
> [3] Harada et al., EMNLP 2025 — Large-scale SFT study shows negative performance depending on data–model mismatch.
> [4] Chatterjee et al., TACL 2025 — Response-only training reduces robustness and generalization on reasoning tasks.
>
> **3. Computational cost analysis**
>
> **Response** Since cost is hardware dependent we analyze the number of tokens used. Our method’s main overhead is one-time offline data construction: generating a set of rewrites O(length of question). Then we sample K=2 traces for the original question and rewrite 6x2|CoT|. Then we structure and create the meta-information for the annotated chains ready for clustering <2|CoT|. If we assume $|question|\approx |CoT|$  we get that the average token cost of our method  is $<20\bullet |CoT|$. In fact we see it is way least. We plot the tokens created by the baseline model and by our model with the relative improvement over the baseline performance. See the cost analysis graph in this [LINK](https://drive.google.com/file/d/1UOL3UcdvazXQUcJkcM8MyOoBK2qyWwe6/view?usp=drive_link).
>
> **4. "...analyze how much the predictor's precision affects ... downstream accuracy"**
>
> **Response** We run an ablation where we use other SBC prediction models as the classifier. We use the next best predictor from our ablation study: Graph Transformer.  We retrain GPT4.1-mini on two datasets with the lowest baseline performance:
>
> | | MMLU | BigMath |
> |--|--|--|
> | GNN(curr) | 78.0 | 74.8 |
> | Graph Transformer | 77.5 | 73.1 |
> We will include this ablation in the appendix of the final version. We observe a similar trend on the rest of the models.
>
> **5. HLE statistical test**
>
> **Response** We ran a McNemar's test on paired per-question outcomes confirming that the improvement is statistically significant (p < 0.05 ; we will add this to the paper). On the broader point about magnitude: on HLE — evaluated on the ~2,000 text-only available samples — a 2.7 percentage point improvement is substantive by the standards of this benchmark, where the gap between top-ranked systems on the public leaderboard is typically 1–2 points.

---

> > ### Author Rebuttal · Reviewer_2daF · 2026-04-03
> >
> > Thank you for the rebuttal.

---

### Official Review · Reviewer_B922 · 2026-03-13

**Soundness:** 3
**Presentation:** 2
**Significance:** 3
**Originality:** 2
**Overall Recommendation:** 4
**Confidence:** 3

**Summary:**

This paper uses semantic preserving examples to train LLMs. The semantic preserving examples and positive/negative pairs are generated by sampling the solution traces from the model with rewrited questions, then the reasoning traces are discretized and annotated to construct the reasoning graph, and define a Solution Boundary Cut (SBC) that separates recoverable states from irrecoverable ones based on whether a correct terminal answer remains reachable in the constructed graph. A boundary-crossing predictor is then trained to identify transitions associated with systematic failure, and use these predicted crossings to get the positive/negative signal used to later DPO style  finetuning or in context learning. Finetuning on four models shows consistent improvement on chosen benchmarks. In addition, when use in context learning with gpt-5.2 on HLE text only they obtained 38.1% performance.

**Compliance With Llm Reviewing Policy:**

Affirmed.

**Final Justification:**

The rebuttal addressed most of my concerns

**Key Questions For Authors:**

see above

**Limitations:**

Yes

**Strengths And Weaknesses:**

Overall, this paper is easy to read and follow. The overall process seems heavily depends on the on the discretization and annotation, I am wondering if different LLM (stronger, weaker) would have a significant impact on this. Additionally, not sure how much the cost of annotation is, it would be great if the authors can share some information on the cost perspective. Also, it would be great to see if fine-tuning or in context learning actually helps to address answer drift with the semantic preserving transformation. The experiments demonstrated that using such augmentation can help models do better in the original questions. It would be great to have such study to direct check the impact.

---

> ### Author Rebuttal · Authors · 2026-03-31
>
> We thank the reviewers for their questions and address each concern below:
>
> **1.***"....it is unclear how much using a stronger or weaker LLM for this step would significantly impact results..."*
>
> **Response:**
> To assess the sensitivity of our method to the LLM used for discretization and annotation, we reran the full pipeline with two different annotators: GPT-5-mini (reported model in the paper) and GPT-4o-mini. We report results for GPT-4.1-mini fine-tuned with our method on MMLU and DROP below:
>
> | | MMLU | DROP |
> |--|--|--|
> | Annotated by GPT-5-mini (our) | 0.87 | 0.91 |
> | Annotated by GPT-4o-mini | 0.92| 0.94|
>
> The results are very similar across annotators, with less than a 1-point drop on both datasets when using the weaker annotation model. This suggests that our pipeline is not highly sensitive to the particular LLM used for discretization and annotation, and that the gains are not driven by a single strong annotator. We observed the same overall trend for the other evaluated models as well. Similar behavior was observed with other models as well.
>
> **2.***"...The annotation/computational cost is never reported..."*
>
> **Response:**
> Our method’s main overhead is one-time offline data construction: generating a set of rewrites O(|question|). Then we sample K=2 traces for the original question and rewrite 5x2|CoT|. Then we structure and create the meta-information for the annotated chains ready for clustering <2|CoT|. Assume O(|question|) = O(|CoT|) we get that the overall the average token cost of our method  is $<12\bullet |CoT|$. We plot the tokens created by the baseline model and by our model with the relative improvement over the baseline performance.
> Please see graph in think [LINK](https://drive.google.com/file/d/1UOL3UcdvazXQUcJkcM8MyOoBK2qyWwe6/view?usp=drive_link).
>
> **3.***"...It would be great to see if fine-tuning or in context learning actually helps to address answer drift with the semantic preserving transformation. The experiments demonstrated that using such augmentation can help models do better in the original questions. It would be great to have such study to direct check the impact..."*
>
> **Response:** We directly evaluated whether fine-tuning improves robustness to semantic-preserving transformations by testing both GPT-4.1-mini and its fine-tuned counterpart on the augmented test split. The average results across all three datasets are shown below.
> | | Normal Dataset| Augmented Dataset|
> |--|--|--|
> | GPT-4.1-mini | 76.5 | 72.0 |
> | FT GPT-4.1-mini | 81.5| 76.1 |
>
> These results show that fine-tuning improves performance not only on the original questions, but also on semantically equivalent transformed versions. In particular, the fine-tuned model consistently outperforms the base model on the augmented test set (+4.1 points), indicating improved robustness to answer drift under semantic-preserving transformations. That said, performance on the augmented set remains lower than on the original set for both models, suggesting that the transformations still introduce a meaningful challenge. We observe the same overall trend for other models as well.

---

> > ### Author Rebuttal · Reviewer_B922 · 2026-04-03
> >
> > thanks for the response

---

### Official Review · Reviewer_gKvY · 2026-03-14

**Soundness:** 3
**Presentation:** 3
**Significance:** 3
**Originality:** 3
**Overall Recommendation:** 5
**Confidence:** 4

**Summary:**

- Meaning-perserving rewrites of a question, such as paraphrases, format changes, and adding benign distractors, LLM answers often change and fail.
- To pinpoint the rewrite types that repeatedly trigger the same error patterns, this paper proposes a procedure to (1) derive a graph representation for a model’s sampled answers to a question and its rewritten versions and (2) identify where the solutions enter regions that reliably end in incorrect answers. To generalize beyond questions, a model-specific boundary-crossing predictor is trained to predict unrecoverable mistakes that will lead to final mistakes.
- To mitigate the problem, it generate a small set of examples to mirror the rewrite patterns that are most responsible for the divergences and use them to steer the model via in-context learning or finetuning. Empirical results on MMLU-Pro, Big-MATH, DROP, and Humanity’s Last Exam show the effectiveness of the steering.

**Compliance With Llm Reviewing Policy:**

Affirmed.

**Final Justification:**

The rebuttal addressed my questions and I believe the paper is solid.

**Key Questions For Authors:**

- L114 left: Why do you need to sample 2 attempts per question? Will the method still work with 1 attempt or 3+ attempts?
- L147 left: What’s the use of slot/variable tags? Are the set of possible tags predefined?
- L119 says “We treat it only as an observable textual artifact of the decoding process and do not assume it is a faithful account of the model’s internal computation.” L160 left says “We emphasize that discretization is used as a post-hoc probe to reverse-engineer model behavior from black-box outputs, rather than as a claim about internal representations.” Could you explain these? What are their implications?
- L146 left: Is there evidence that slot/variable tags and operation labels are necessary and beneficial for merging graph nodes?
- L157 right “To avoid quadratic comparisons, we use approximate nearest neighbor search over step embeddings for node lookup”: What are you comparing and why are you doing node lookup? Is this about node merging?

**Limitations:**

yes

**Strengths And Weaknesses:**

- The claims in the paper are generally supported by empirical evidence. The proposed method is effective on 5 recent LLMs on 4 tasks. Comprehensive baseline evaluation and ablation are provided.
- The paper is well-written and very easy to follow. The running examples and definitions are all very helpful.
- The paper identifies a neat problem—LLMs are not robust to meaning-perserving rewrites of questions—and advances the understanding and mitigation of the problem.
- The findings are novel and method is a somewhat novel combination of various existing methods.
- There is still room for polishing the paper (see Questions). But I did not observe outstanding weaknesses.

---

> ### Author Rebuttal · Authors · 2026-03-31
>
> We thank the reviewer for their constructive feedback, please find our clarification to the proposed questions:
>
> **1. L114 left: Why do you need to sample 2 attempts per question? Will the method still work with 1 attempt or 3+ attempts?**
>
> **Response:** Unrecoverability is defined with respect to the constructed graph and may be affected by sampling, as discussed in Sec. 4.1. To evaluate this sensitivity, we include here an ablation varying the number of sampled trajectories per rewrite (K=1, 2, 3, 5, 10) using GPT-4.1-mini, resulting in over 50 traces per question (5 question augmentations x K). See the output graph in this [link](https://drive.google.com/file/d/12YycbpJlwL66fQlbBTB6f2iKmZJi57hL/view?usp=sharing).
> In practice, we select K=2 as a sweet spot between computational efficiency and boundary estimation accuracy. More exhaustive sampling is unlikely to materially change the learned boundary, and exploring higher-K regimes at scale remains future work. We will include this analysis in the appendix and clarify this point in the final version. We observe a similar trend for other models as well.
>
> **2.L147 left: What’s the use of slot/variable tags? Are the set of possible tags predefined?**
>
> **Response:** The slot/variable tags serve as semantic anchors that reduce reliance on the specific surface text of each step, improving the precision of the node-merging step: each step is represented by a weighted combination of its text embedding and its tag embedding, and both are used to decide whether two steps across traces map to the same graph node (Section 3.3). The slot/variable tags are generated freely by GPT-5-MINI conditioned on the question and trace, while the operation labels (RETRIEVE, INFER, GENERATED, OTHER) are drawn from a predefined set reflecting the possible reasoning operations an LLM can perform (Section 3.2). We will clarify this distinction in the revision.
>
> **3. L119 says “We treat it only as an observable textual artifact of the decoding process and do not assume it is a faithful account of the model’s internal computation.” L160 left says “We emphasize that discretization is used as a post-hoc probe to reverse-engineer model behavior from black-box outputs, rather than as a claim about internal representations.” Could you explain these? What are their implications?**
>
> **Response** Both statements clarify the scope of the method. The “solution traces” we analyze are generated text, i.e., outputs of the decoding process, not direct access to the model’s internal reasoning, or abstract ‘thinking steps’.
> Accordingly, the discretization step should be understood as a post-hoc analysis tool applied to these outputs. It allows us to structure and compare trajectories across runs, but it does not claim that the underlying model internally operates over discrete steps or graph-like states nor that this is the steps clustering that it uses.
>  We claim that the reasoning graph is an empirical behavioral abstraction of the model under stochastic decoding, useful for identifying recurring failure patterns, rather than a mechanistic interpretation of the model’s internal representations.
>
> **4.L146 left: Is there evidence that slot/variable tags and operation labels are necessary and beneficial for merging graph nodes?**
>
> **Response** To directly test whether slot/variable tags and operation labels are useful for node merging, we performed an ablation in which we remove each component and measure merge quality.
> We evaluate merge quality using an LLM-as-a-judge audit with GPT-5. For each candidate merge, the judge is given the original question, the reasoning chains being merged, and the resulting merged node, and returns a binary decision indicating whether the merge is semantically valid.
>
> | | Without Variable | With Variable|
> |--|--|--|
> | With Op. Label | 0.87 | 0.91 |
> | Without Op. Label | 0.92| 0.94|
>
> **5. L157 right “To avoid quadratic comparisons, we use approximate nearest neighbor search over step embeddings for node lookup”: What are you comparing and why are you doing node lookup? Is this about node merging?**
>
> **Response** Yes, this refers to node merging. Each new step is compared to existing nodes to decide whether to merge it or create a new node. Instead of comparing against all nodes (quadratic cost), we use approximate nearest neighbor search to retrieve only the most similar candidates for this lookup.

---

> > ### Author Rebuttal · Reviewer_gKvY · 2026-03-31
> >
> > - For 4, is it true that actually Without Op. Label has higher performance according to the new result? What do "Without Variable" and "With Variable" mean? How about with and without slot tags?
> > - For 5, what's the exact algorithm for your approximate nearest neighbor search?

---

> > > ### Author Response · Authors · 2026-04-06
> > >
> > > 1. For (4), thank you for pointing this out. There is a typo in the table: the labels “With Op. Label” and “Without Op. Label” were accidentally swapped. After correction, using operation labels does improve merge quality, as intended. We will fix this in the final version. To clarify the terminology, “Variable” in the table refers to the slot/variable tags described in L147–157, namely semantic tags indicating which quantity or entity a step concerns, while the operation labels are the coarse step types such as RETRIEVE, INFER, GENERATED, and OTHER. The main finding is that both signals help, and the best merge quality is obtained when both are included. We will revise the wording and table to make this explicit.
> > >
> > > 2. For (5), we use HNSW (Hierarchical Navigable Small World) as the approximate nearest neighbor search algorithm for node lookup during graph construction. We will state this explicitly in the final version.

---

### Decision · Program_Chairs · 2026-04-30

**Decision:**

Accept (regular)

**Comment:**

This paper addresses the instability of LLMs under meaning-preserving rewrites by proposing Targeted Alternation Training. The method identifies specific divergence points where reasoning collapses across semantically equivalent inputs and uses these to steer the model toward robustness. All reviewers are positive. After considering the reviews and rebuttal, I recommend this paper for Acceptance.